# Advancing Precipitation Prediction Using a New Generation Storm-resolving Model Framework - SIMA-MPAS (V1.0): a Case Study over the Western United States

Xingying Huang*, Andrew Gettelman, William C. Skamarock, Peter Hjort Lauritzen, Miles Curry, Adam Herrington, John T. Truesdale, Michael Duda

Affiliation: National Center for Atmospheric Research, Boulder, CO 80305, USA

*Correspondence to*: Xingying Huang (xyhuang@ucar.edu)

**Abstract:** Global climate models (GCMs) have advanced in many ways as computing power has allowed more complexity and finer resolution. As GCMs reach storm-resolving scale, they need to be able to produce realistic precipitation intensity, duration, and frequency at fine scales with consideration of scale-aware parameterization. This study uses a state-of-art storm-resolving GCM with a nonhydrostatic dynamical core - the Model for Prediction Across Scales (MPAS), incorporated in the atmospheric component (Community Atmosphere Model, CAM) of the open-source Community Earth System Model (CESM), within the System for Integrated Modeling of the Atmosphere (SIMA) framework. At uniform coarse (here, at 120km) grid resolution, the SIMA-MPAS configuration is comparable to the standard hydrostatic CESM (with finite-volume (FV) dynamical core) with reasonable energy and mass conservation on climatological timescales. With the comparable energy and mass balance performance between CAM-FV (workhorse dycore) and SIMA-MPAS (newly developed dycore), it gives confidence in SIMA-MPAS's applications at a finer resolution. To evaluate this, we focus on how the SIMA-MPAS model performs when reaching storm-resolving scale at 3km. To do this efficiently, we compose a case study using a SIMA-MPAS variable resolution configuration with a refined mesh of 3km covering the western US and 60 km over the rest of the globe. We evaluated the model performance using satellite and station-based gridded observations with comparison to a traditional regional climate model (WRF, the Weather Research and Forecasting model). Our results show realistic representations of precipitation over the refined complex terrains temporally and spatially. Along with much improved near-surface temperature, realistic topography and land-air interactions, we also demonstrate significantly enhanced snowpack distributions. This work illustrates that a global SIMA-MPAS at storm-resolving resolution can produce much more realistic regional climate variability, fine-scale features, and extremes to advance both climate and weather studies. This next-generation storm-resolving model could ultimately bridge large-scale forcing constraints and better-informed climate impacts and weather predictions across scales.

## 1 Introduction

Climate models have advanced in many ways in the last decade including their atmospheric dynamical core and parameterization components. Advances in computer power have now enabled climate models to be run with non-hydrostatic dynamical cores at "storm-resolving" scales, on the order of a few kilometers (Satoh et al., 2019). These GSRMs (Global Storm-Resolving Models) have been constructed at a number of modeling centers (Satoh et al., 2019; Stevens et al., 2019; Dueben et al., 2020, Stevens et al., 2020, Caldwell et al., 2021). We expect an emerging trend in improving and applying the new modeling structures for a better and more accurate understanding of global and regional climate studies and weather-scale predictions.

The Community Earth System Model (CESM) has been used in a wide range of climate studies. For high-resolution CESM applications (but hydrostatic only), variable-resolution (VR) CESM-SE (spectral element core) for regional climate modeling has been used in many regional climate studies (such as Small et al., 2014; Zarzycki et al., 2014, 2015; Rhoades et al., 2016; Huang et al., 2016, 2017; Bacmeister et al., 2018; Gettelman et al., 2018, 2019; Van et al., 2019). Specifically, Rhoades et al. (2016) found that the VR-CESM framework (with refinement at 0.25° and 0.125° resolutions) can provide much enhanced representation of snowpack properties relative to widely used GCMs (such as CESM-FV 1° and CESM-FV 0.25°) over the California region. Gettelman et al. (2018) found that the variable-resolution CESM-SE simulation (at 0.25°, ~25 km) can produce precipitation intensities similar to the high-resolution, and has higher extreme precipitation frequency than the low-resolution simulation over the Continental United States (CONUS) refinement region, close to observations.

More recently for storm-resolving model development, there have been two efforts to bring the dynamical core from the Model for Prediction Across Scales (MPAS) into CESM. The first effort involved implementing the hydrostatic atmospheric dynamical core in MPAS Version 1 in the Community Atmosphere Model (CAM), which is the atmospheric component of CESM. This effort made available the horizontal variable-resolution mesh capability of the MPAS spherical centroidal Voronoi mesh (Ringler et al., 2010), and led to a number of studies (e.g., Rauscher et al., 2013; Rauscher & Ringler, 2014; Sakaguchi et al., 2016). For example, Rauscher et al. (2013) found that tropical precipitation increases with increasing resolution in the CAM-MPAS using aquaplanet simulations.

Later, the static port of MPAS to CAM was updated with the nonhydrostatic MPAS atmospheric solver (Skamarock et al., 2012; Skamarock et al., 2014) to provide nonhydrostatic GSRM capabilities to CAM (Zhao et al., 2016). Neither of these ports was formally released, and the nonhydrostatic MPAS was not energetically consistent with CAM physics, or its energy fixer

given, among other things, the height vertical coordinate used by MPAS. Furthermore, the MPAS modeling system and its dynamical core, being separate from CESM, have evolved from these earlier ports. To address the issues in the earlier MPAS dynamical core ports to CAM/CESM, the MPAS nonhydrostatic dynamical core has been brought into CAM/CESM as an external component, i.e., it is pulled from the MPAS development repository when CAM is built, and all advances in MPAS are immediately available to CESM-based configurations using MPAS. This latest port was accomplished as part of the SIMA (System for Integrated Modeling of the Atmosphere) project. Importantly, this implementation also includes an energetically consistent configuration of MPAS, with its height vertical coordinate, the CAM hydrostatic-pressure coordinate physics and the CAM energy fixer.

The MPAS dynamical core solves the fully compressible nonhydrostatic equations of motion and continues to be developed and used in many studies (Feng et al., 2021; Lin et al., 2022; also see https://mpas-dev.github.io/atmosphere/atmosphere.html). In this work, we test the storm-resolving capabilities in this new atmospheric simulation system. We use SIMA capabilities to configure a version of CESM with the MPAS nonhydrostatic dynamical core, called SIMA-MPAS instead of CESM-MPAS, since it is coupled only to a land model, with the other climate-system components being data components. In particular, we would like to answer the question: can a nonhydrostatic dycore coupled global climate model reproduce observed wet season precipitation over targeted refinement regions? In addition, will this new development and modeling framework perform better or worse than a mesoscale model at similar resolution?

We aim to understand how this new SIMA-MPAS model configuration performs when configured for storm-resolving (convection-permitting) scale for precipitation prediction over the western United States (WUS). Leveraging the recent significant progress in SIMA-MPAS development, we have undertaken experiments to understand the performance of SIMA-MPAS in precipitation simulations involving heavy storm events and relevant hydroclimate features at fine scales. We also explore large-scale dynamics and moisture flux transport over the subtropical region across the North Pacific. We evaluate the model results compared to both observations and a regional climate model. Employing the recent modeling developments in CESM with the MPAS dycore, the ultimate goal of this study is to evaluate the potential improvements to our understanding of atmospheric processes and predictions made possible with GSRM capabilities. We begin in section 2 with a description of the model configurations and experiments. Section 3 describes the main results, including mean climatology diagnostics, precipitation and snowpack statistics, and large-scale moisture flux and dynamics. A summary and discussion follow in Section 4.

## 2 Methods, experiments, and dataset

### 2.1 Methods and experiments

As briefly mentioned in the introduction section, we configure CESM2 (Danabasoglu et al., 2020) with the MPAS nonhydrostatic dynamical core and CAM6 physics. We call this configuration SIMA-MPAS. SIMA is a flexible system for configuring atmospheric models inside of an Earth System Model for climate, weather, chemistry and geospace applications (https://sima.ucar.edu). The components of this particular configuration also include the coupled land model CLM5 (with MOSART river model) and prescribed observation-based SST (sea surface temperature) and ice. MPAS-Atmosphere employs a horizontal unstructured centroidal Voronoi tessellation (CVT) with a C-grid staggering (Ringler et al., 2010), and its numerics exactly conserve mass and scalar mass. Both horizontal uniform meshes and variable resolution meshes with smooth resolution transitions are available for MPAS-Atmosphere, and this study employs both mesh types. It uses a hybrid terrain-following height coordinate (Klemp 2011).

We summarize here the key developments on the coupling of MPAS dynamical core to CAM physics and changes to CAM physics to accommodate MPAS. Most of all, we would like to point out that a consistent coupling of the MPAS dynamic core with the CAM physics package is not trivial for several reasons. 1) MPAS uses a height (z) based vertical coordinate whereas CAM physics uses pressure. 2) The CAM physics package enforces energy conservation by requiring each parameterization to have a closed energy budget under the constant pressure assumption (Lauritzen et al., 2022). For the physics-dynamics coupling to be energy consistent (i.e., not be a spurious source/sink of energy) requires the energy increments in physics to match the energy increments in the dynamical core when adding the physics tendencies to the dynamics state. When "mixing" two vertical coordinates, that becomes non-trivial. 3) The prognostic state in MPAS is based on a modified potential temperature, density, winds, and dry mixing ratios whereas CAM uses temperature, pressure, winds and moist mixing ratios for the water species. The conversion between (discrete) prognostic states should not be a spurious source/sink of energy either. 4) Lastly, the energy fixer in CAM that restores energy conservation due to updating pressure (based on water leaving/entering the column), as well as energy dissipation in the dynamical core and physics-dynamics coupling errors (see Lauritzen and Williamson, 2019), assumes a constant pressure upper boundary condition. MPAS assumes constant height at the model top, so the energy fixer needs to use an energy formula consistent with the constant volume assumption. The details of the energy consistent physics-dynamics coupling and extensive modifications to CAM physics to accommodate MPAS are beyond the scope of this paper and will be documented in a separate source.

In terms of scale awareness, there are two aspects related to the model physics in the configuration that must be considered when employing regionally refined meshes. First, features resolvable in the finer regions of the mesh may not be resolvable in the coarser regions of the mesh. These features, e.g. deep convection in this study, need to be parameterized in the coarse mesh regions and not parameterized in the fine mesh regions, typically with the parameterization reducing its adjustment gradually in the mesh transition regions. Second, the timestep used for the physics is the same over the entire mesh. i.e. in both coarse and fine regions, and the timestep in CESM-

MPAS is chosen to be appropriate for the smallest grid, as indicated in Table 1. Within our simulations, the balance of deep convective (diagnostic) and stratiform (large-scale) precipitation changes with the mesh spacing. In addition, since the deep convective parameterization in CESM-MPAS has a closure with a fixed timescale, the parameterized convection produces less condensation in the coarse mesh regions compared to simulations with a larger timestep appropriate for the coarser mesh (Gettelman et al 2019). But in the simulations herein, most of the precipitation is strongly forced by the large-scale flow, with the larger condensation hypothesized to lead to larger rain rates. This is particularly important over the WUS complex terrains. The large scale condensation scheme, part of the unified turbulence scheme (Golaz et al., 2002) has internal length scales that should adjust its distributions as the scale changes (less variance in the PDFs). Land surface related feedback is also resolution dependent with scale-aware surface heterogeneity and coupled land-atmosphere interactions to affect the phase and hydrological impacts resulting from the regional precipitation statistics.

With the above significant progress in SIMA-MPAS development, we would like to diagnose the performance of this new generation model when applied at convection-permitting resolutions and when bridging both weather and climate scale simulations in a single global model. We have chosen the WUS (due to its hydroclimate vulnerability and complexity, heavily impacted by precipitation variability) as our study region to examine the precipitation features in SIMA-MPAS at fine scales during wet seasons. We aim to figure out when the model outperforms and underperforms when compared with a traditional regional climate model against best-available observations and observationally based gridded products at similar resolutions for mean and extreme precipitation. As mentioned in the introduction, we would like to figure out whether a nonhydrostatic dycore coupled global climate model can reproduce observed wet season precipitation over targeted refinement regions with heavy impacts. And will this new development and modeling framework perform better or worse than a mesoscale model at similar resolution? Those are important questions to answer given the long-standing biases in traditional hydrostatic GCMs for simulating heavy precipitation and extremes.

To answer those questions, we have designed and conducted a set of experiments as shown in Table 1. In detail:

- Set A: We have tested CESM2 at the same coarse resolution using both MPAS (at 120km) as the nonhydrostatic core and Finite Volume (Danabasoglu et al., 2020) (at ~1 degree) as the hydrostatic core for multiple years of climatology to get five-year mean F2000 climatology (in which, the SST and ice condition are prescribed at the same yearly climatology with mean from the time period 1995-2005) at ~1° for both MPAS and FV (finite-volume) dycore.

- Set B: As the main focus for this work, a variable resolution mesh is configured with 3km refinement centered over WUS as shown in Figure 1, for five wet-season simulations with 60-3km mesh (years 1999 to 2004; mid-November to mid-March; FHIST component set for historical forcings); atmosphere conditions initialized by Climate Forecast System Reanalysis (CFSR) data.

- Set C: In addition, we have also configured uniform 60km simulations for two wet seasons in contrast to the 60-3km ones (years 2000 to 2002; November to March).

- Set D: Lastly, to accommodate the recent changes to the MG microphysics scheme, we have also conducted simulations at 60-3km resolution for the three wet seasons (years 1999-2002) using MG3 with graupel (Gettelman et al., 2019) instead of MG2 (Gettelman and Morrison 2015) as in the Set B simulations. Specifically, Gettelman et al 2019 (i.e., the MG3 paper) show that even at 14 km scale the inclusion of rimed ice changes the timing and location of precipitation in the Western United States due to the different fall speeds and lifetimes of graupel, which is formed when higher vertical velocities result. This effect is expected to be larger at 3km.

All simulations have been conducted with 58 vertical levels up to 43 km. Set A also includes experiments using 32 vertical levels. We have used the default radiation time step (1 hour). The physics and dynamic timesteps are set to default at 1800s for ~1° degree CAM-FV simulation, and this is the default for CAM6 physics for the nominally 1 degree. For 120km the MPAS dynamic timestep is 900s and the physics timestep is 1800s. We also use 900s for the 60km grid-space experiments, scaling it with reduced mesh spacing. The dynamic time-step for MPAS dycore is 20s for 60-3km experiments with physics time-step set to 120s. Instead of using a 20s timestep for the 60-3 km mesh as scaling would imply, we use a 120s physics timestep for the 60-3km experiments, in part to reduce computational cost and because other studies have shown acceptable results with this physics timestep at comparable mesh spacing (e.g., Zeman et al 2021). We also recognize that the WUS precipitation as the focus of our study is predominantly orographically forced, whereas the physics-timestep-critical processes are related to unstable deep convection, perhaps lending support for a longer physics timestep in this application. We acknowledge the possible sensitivity of our results to the physics timestep and we will be examining this more in future work. The average cost for 60-3km simulations including writes and restarts is ~4K to 6K core-hour for one-day simulation (i.e., ~120K to 180K for getting 30-day output) using the Cheyenne supercomputer with the scaling of the high-performance computing to be further improved. We would like to acknowledge that model tuning is not performed. Given the interannual variability of precipitation over the WUS study region, we also acknowledge that it is not our goal to reproduce the recent historical climatology but to evaluate the overall model performance.

**Table 1: A list of experiments in this study and the key configuration information**

| Dycore/Model experiments | Component set | Grid spacing | Grid columns | Simulation time | Vertical level | Physics/dynamics timestep and microphysics |
|---|---|---|---|---|---|---|
| MPAS | F2000climo | 120km | 40962 | 5 years | 32L, 58L | 1800s/900s, MG2 |
| FV | F2000climo | ~1degree | 55296 | 5 years | 32L, 58L | 1800s/1800s, MG2 |
| MPAS | FHIST | 60-3km | 835586 | 1999-2004, Nov. - March | 58L | 120s/20s, MG2 |
| MPAS | FHIST | 60-3km | 835586 | 1999-2000, Nov. - March | 58L | 120s/20s, MG3 |
| MPAS | FHIST | 60km | 163842 | 2000-2002 | 58L | 900s/450s, MG2 |

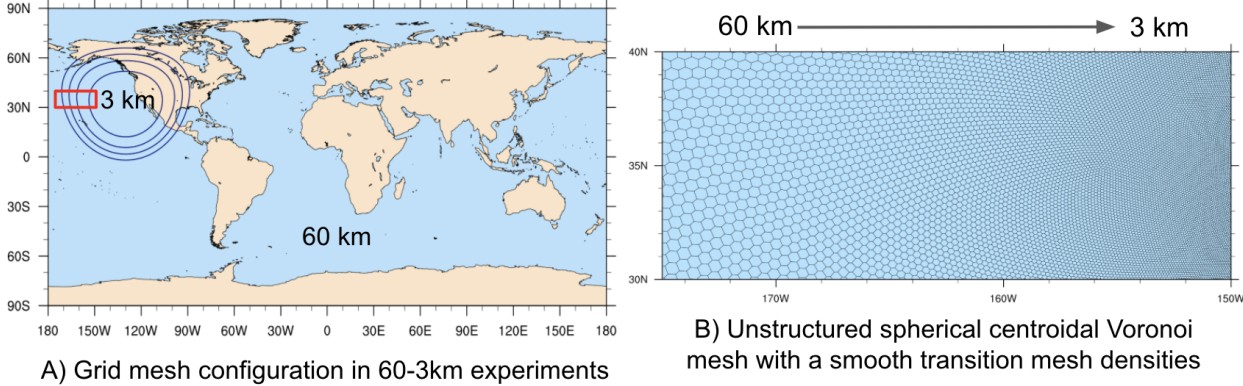

A) Grid mesh configuration in 60-3km experiments

B) Unstructured spherical centroidal Voronoi mesh with a smooth transition mesh densities

**Figure 1:** SIMA-MPAS mesh configuration for the 60-3km experiments**.** A) The global domain mesh configuration with total grid columns of 835586; B) The zoomed-in region (see the red box depicted in panel A)) for the mesh structure from 60km to 3km.

## 2.2 Observations and observationally-based gridded products used to evaluate model performance

In this work, we have employed observations from CERES EBAF products (Kato et al., 2018; Loeb et al., 2018) for cloud and radiation fluxes properties. We have used GHCN Gridded V2 data (Fan and Van, 2008) for the land 2m air temperature globally, which is provided by the NOAA/OAR/ESRL PSL. We have also used PRISM data for gridded observed precipitation and temperature features (Daly et al., 2017) and gridded 4 km observational data for snow water

equivalent (Zeng et al., 2018). We have also used the recently released Livneh precipitation data (Pierce et al., 2021) as another gridded observationally-based precipitation dataset to better account for extreme precipitation. Another important dataset used for comparison is the WRF (Weather Research and Forecasting) model 4km simulation data over CONUS from Rasmussen et al. (2021, https://rda.ucar.edu/datasets/ds612.5), which used the mean of the CMIP5 model as the boundary forcing. We extracted the same historical time data as the 60-3km simulations for direct evaluation (i.e., nonhydrostatic CESM vs. nonhydrostatic WRF as a widely used regional climate model).

Detailed descriptions of the open-shared datasets used in this study are given below:

- CERES EBAF data products: we use gridded data from the Energy Balance And Filled (EBAF) product from the NASA Clouds in the Earth's Radiant Energy System (CERES), described by Loeb et al (2018). CERES provides high quality top of the atmosphere radiative fluxes and cloud radiative effects, as well as consistent ancillary products for Liquid Water Path (LWP) and cloud fraction. We start with monthly mean gridded products at 1° and make a 20 year climatology from 2000-2020.

- GHCN_CAMS Gridded 2m air land temperature: global analysis monthly data from NOAA PSL comes with resolution at 0.5 x 0.5°. It combines two large networks of station observations including the GHCN (Global Historical Climatology Network version 2) and the CAMS (Climate Anomaly Monitoring System), together with some unique interpolation methods (*https://psl.noaa.gov;* Fan and Van, 2008).

- PRISM observed data: the Parameter-elevation Regressions on Independent Slopes Model (PRISM) gridded observed data for daily precipitation and daily 2m air temperature is used at 4 km grid resolution (Daly et al., 2017; https://prism.oregonstate.edu/). Covering Continental U.S., PRISM takes the station observations from the Global Historical Climatology Network Daily (GHCND) data set (Menne et al., 2012) and applies a weighted regression scheme that accounts for multiple factors affecting the local climatology (Daly et al., 2017).

- Livneh gridded observationally-based precipitation dataset: in addition to PRISM data, to better account for extreme precipitation, a recently released Livneh precipitation data (Pierce et al., 2021; http://cirrus.ucsd.edu/~pierce/nonsplit_precip/) is also used for model evaluation. The data (~6km grid resolution) is shown to perform significantly better in reproducing extreme precipitation metrics (Pierce et al., 2021).

- Snow water equivalent (SWE) data over the CONUS: this is the observational data product we use for snowpack diagnostics. The data is available from National Snow and Ice Data Center (NSIDC) (at https://nsidc.org/data/nsidc-0719/versions/1). The product provides

daily 4km SWE from 1981 to 2021, developed at the University of Arizona. The data assimilated in-situ snow measurements from the SNOTEL network and the COOP network with modeled, gridded temperature and precipitation data from PRISM (Zeng et al., 2018; Broxton et al., 2019).

- CONUS (Continental U.S.) II high resolution climate simulations: The WRF (Weather Research and Forecasting) nonhydrostatic model simulations we used for comparison are from Rasmussen et al. (2021) (accessible at https://rda.ucar.edu/datasets/ds612.5). Its horizontal grid resolution is 4 km with forcing from the mean of the CMIP5 model for both present (1996-2015) and future (2080-2099) mean climate, with hourly output. For the study region we focus on here (i.e., over the western US), the simulations provide a more realistic depiction of the mesoscale terrain features, critical to the successful simulation of mountainous precipitation (Rasmussen et al., 2021).

The topography details are shown in Figure 2 over the western US study region, showing that the complex terrains over coastal and mountainous regions have been well-resolved in SIMA-MPAS at 3 km resolution (in contrast to 60 km). This is comparable to the topography details in the WRF mesoscale model at a similar resolution. We do notice the smoother topography in SIMA-MPAS over the 3km mesh bounds and transient domains (see Figure S1). For future regional refined applications, we would suggest having a reasonably larger domain area than the study region at the finest resolution to accommodate the noise and instability from mesh transition. When applied, we regridded the SIMA-MPAS model data to the same grid resolution as the PRISM observation and WRF reference data (i.e., 4 km). For the regridded method and procedure, first CAM-MPAS data is remapped from unstructured grids to regular rectilinear lat/lon grids at 0.03 degree, and then the rectilinear data is regridded to the same grid spacings as the PRISM using the bilinear interpolation. The orographic gravity wave drag scheme in SIMA-MPAS (used in CESM2-CAM6) uses a 'sub-grid' orography to force the scheme. Sub-grid orography is calculated for each grid cell from a standard high resolution (1km) Digital Elevation Model. Thus, the sub-grid orography forcing is small at 3km, and is larger at 60km, and varies with grid cell size. So, the overall drag should be somewhat similar to the scale, but partitioned differently between resolved and unresolved scales.

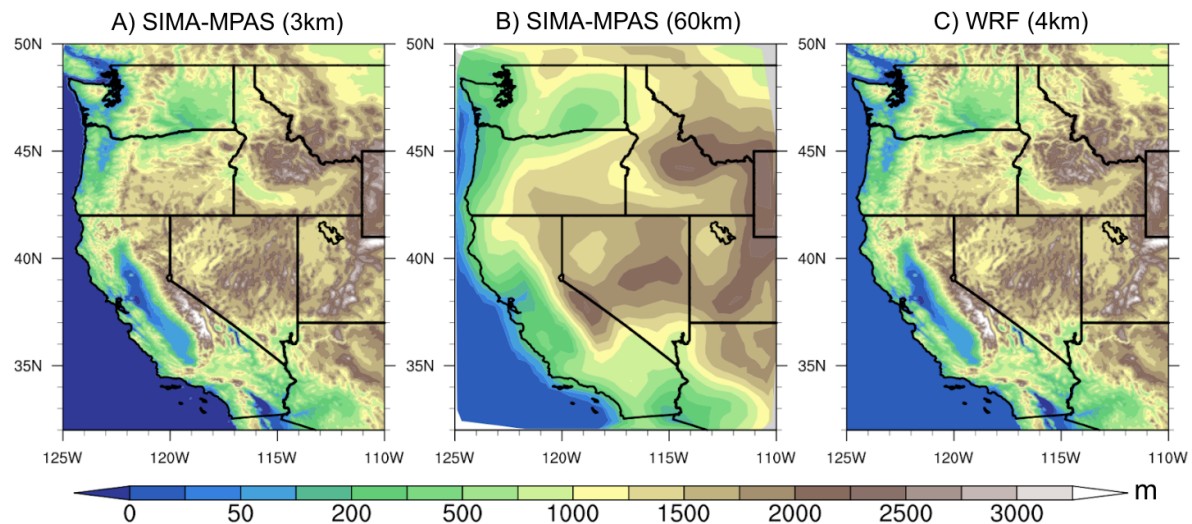

**Figure 2: Topography over the western US region.** A) SIMA-MPAS at 3km refinement, B) SIMA-MPAS uniform 60km grid mesh, and C) WRF simulations at 4km over CONUS.

## 3 Results

### 3.1 Mean climatology diagnostics for CESM with MPAS dycore

As the nonhydrostatic dynamical core is coupled to the CESM model framework, we would like to understand the mean climate in SIMA-MPAS and how that compares to a standard hydrostatic core (here, using FV), with the experiments described in Table 1. We evaluate the global context of the new formulation of CESM with a nonhydrostatic dynamical core with both 32 and 58 vertical levels. The 58 layer has higher resolution in the Planetary Boundary Layer (PBL) and in the mid and upper troposphere (about 10 additional levels in the PBL and decreasing vertical grid spacing from 1000m to ~500m near the tropopause). Satellite observations are used for comparison as described in the above section 2.2. Simulation results are averaged over the five years output under the present-day climatology (with SST and ice forcings from the mean of the period 1996-2005). That means that simulations are forced with the same climatological monthly mean boundary conditions for sea surface temperature and greenhouse gasses every year to reduce interannual variability.

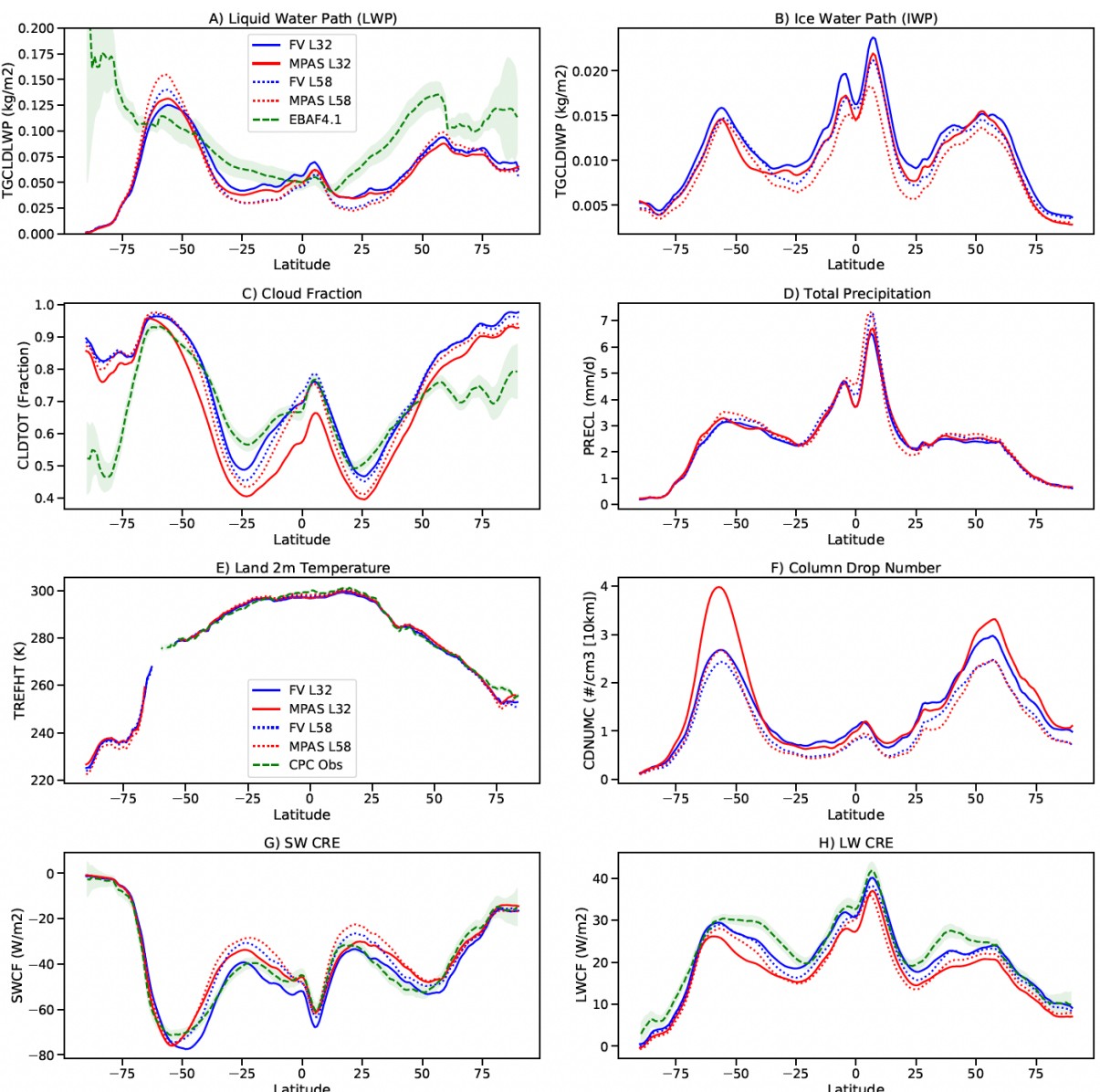

**Figure 3: Zonal mean climatology from 5-year simulations with CESM2 and CAM6 physics using different dynamical cores and vertical levels.** A) Liquid Water Path (LWP), B) Ice Water Path (IWP), C) Cloud Fraction, D) Total precipitation rate, E) Land 2m air Temperature, F) Column drop number, G) Shortwave Cloud Radiative Effect (SW CRE), H) Longwave (LW) CRE. Simulations are the default Finite Volume (FV) dynamical core with 32 levels (FV L32: Blue Solid) and 58 levels (FV L58: Blue Dashed). Also, the MPAS dynamical core with 32 levels (MPAS L32: Red Solid) and 58 levels (MPAS L58). Observations are shown in green for CERES 20-year climatology (from 2000-2020) for LWP, Cloud Fraction, SW CRE, and LW CRE, and GHCN_CAMS Gridded land 2m air temperature from 1990-2010 for E). Shaded values are one sigma annual standard deviations.

Figure 3 indicates that MPAS simulations have a very similar climate to FV simulations. There are some differences in tropical ice water path in the southern hemisphere tropics, and some significant differences in sub-tropical cloud fraction. The climate differences between 32 and 58 levels are also similar between dynamical cores: decreases in liquid and ice water path at higher vertical resolution. SIMA-MPAS has slight increases in cloud fraction and precipitation at higher vertical resolution, while SIMA-FV has little change or slight decreases in cloud fraction. Land surface temperature is well reproduced when ocean temperatures are fixed with both dynamical cores. Column drop number with CAM-MPAS is lower than CAM-FV, but more stable with respect to resolution changes. Subtropical SW CRE and LW CRE have higher magnitudes with CAM-MPAS, consistent with higher LWP and cloud fraction in these regions, yielding better agreement with the meridional CRE structure. When examining the spatial differences (Figure S2 and Figure S3), we further found that the differences in the wind over the oceans drive differences in aerosols (sea salt) which alter the aerosol optical depth and droplet concentration. The radiative effects come as a result of cloud fraction changes: high clouds and specifically ice water path for the longwave, low cloud and liquid Water Path for the shortwave. The signal in clouds is stronger at L32 (Figure 3, Figure S2), again, probably due to larger differences in the PBL, which is better resolved at L58 (Figure 3, Figure S3). The microphysics is not as directly related to the cloud fraction, which means interaction with the boundary layer turbulence is important. While these changes are easy to spot, they are not that large, and generally well within some of the tuning which is often done during the model development process.

Analysis of the atmospheric wind and temperature structure (Figure S4 and Figure S5) indicates that SIMA-MPAS compares as well or better to reanalysis winds and thermal structure in the vertical as SIMA-FV, though biases are different and of a different sign in many regions of the middle atmosphere. There are differences in low level wind speed and the subtropical jets between MPAS and FV (Figure S4), driving differences in temperature between them (Figure S5), particularly in the stratosphere and near the south pole. The stratosphere and free troposphere winds differences are due to slightly different damping and deposition of gravity wave drag forcing. The temperature changes above the surface respond to those wind changes. The near-surface temperature differences (e.g., around Antarctica) also relate to transport of air around topography which is different between MPAS and FV.

Overall, SIMA-MPAS produces a reasonable climate simulation, with biases relative to observations that are of similar magnitude as SIMA-FV simulations, despite limited adjustments being made to momentum forcing. SIMA-MPAS has a realistic zonal wind structure with sub-tropical tropospheric and polar stratospheric jets. There are differences in magnitude from ERAI, but MPAS (which has not been fully tuned) produces a realistic wind distribution. Further tuning of momentum in the dynamical core and physics could reduce these biases. The key feature of this work is that biases in the Northern Hemisphere mid-latitude tropospheric winds are very small for both FV and MPAS. For the temperature profile, there are patterns of bias between the high and low latitudes indicating different stratospheric circulations between the model and the reanalysis. That could be adjusted with the drag and momentum forcing in the model. Note that no adjustment of the physics has been performed.

## 3.2 Precipitation distribution and statistics

### 3.2.1 Mean precipitation features

In the western US during the wet seasons, most of the precipitation occurs over the mountainous regions, with significant impacts on both water resources and potential flood risk management (Hamlet and Lettenmaier, 2007; Dettinger et al., 2011; Huang et al., 2020a). In Figure 4, we show the wet season mean (mid-Nov to mid-Mar as investigated here) precipitation features over the targeted region with differences from observations. Although the observational differences between PRISM and Livneh on average is small, it provides a more robust evaluation for both mean and extreme precipitation by having those two observational products. The result demonstrates that SIMA-MPAS can well simulate the precipitation intensity and spatial distributions, as compared to PRISM and Livneh observations. The spatial features at 3km are well captured with the spatial correlation of about 0.93 with precipitation mainly distributed over the Cascade Range, Coastal Range, Sierra Nevada, and the Rocky Mountains. If looking at the precipitation at the coarser resolution (60km, Figure S6a) in SIMA-MPAS, the mean domain average of the precipitation (~2.43 mm, when averaged over years 2000-2002) is similar to the fine resolution results (~2.61 mm) but lacking important regional variability and spatial details.

In terms of biases when compared to PRISM data, SIMA-MPAS 3km overall underestimates the precipitation by about 0.07 mm (bias averaged over the plotted domain), especially over the windward regions, which could relate to the bias in heavy precipitation frequency and/or the discrepancies in ARs landfalling locations and magnitude from what was observed over the five-year (wet-season) simulation statistics. We acknowledge that the interannual variability and the sample size of the ARs could also affect the results of landfalling precipitation. WRF, on the other hand, tends to overestimate the precipitation in most regions (for about 0.53 mm, bias averaged over the plotted domain compared to PRISM) except for the northwest coast and some Rocky Mountains regions, which can be seen from the relative difference plot (Figure 4c). The relative

differences in precipitation are generally large over the dryer regions in SIMA-MPAS. Overall, compared to PRISM, the bias is negative (for about -0.81 mm on average) over windward regions, but positive over the lee side (for about 0.48 mm on average). We also notice that the spatial details of the precipitation are relatively smoothed over the Rocky Mountains resulting in a large underestimation bias, which could be partly due to the fact that the boundary for the 3km mesh grids is nearing those regions (see Figure 1, Figure 2, and Figure S1).

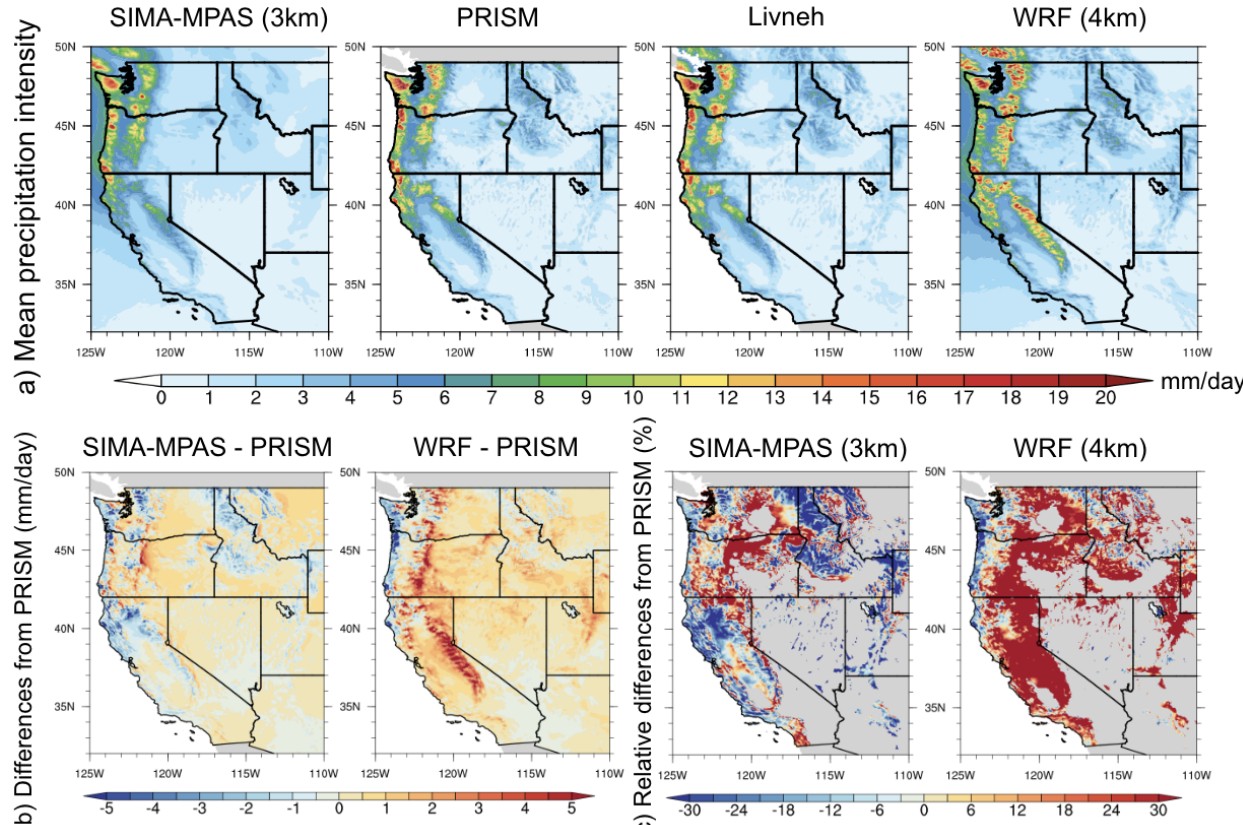

**Figure 4: Mean simulated precipitation and differences from observation:** a) Wet-season (mid-Nov to mid-March) daily precipitation intensity over western US (1999-2004); b) Absolute differences from PRISM reference; c) Similar as b, but for relative differences from PRISM (grid box values less than 1mm/day have been masked)) with the SIMA-MPAS model data regridded to the same resolution as the PRISM grid spacings (i.e., 4 km).

Over the western US, especially in the coastal States, heavy precipitation can be induced by extreme storm events mainly in the form of atmospheric rivers (Leung and Qian, 2009; Neiman et al., 2011; Rutz et al., 2014; Ralph et al., 2019; Huang et al., 2020b). The capability to capture and predict such extreme events is a significant part of the application of weather and climate models (Meehl et al., 2000; Sillmann et al., 2017; Bellprat et al., 2019). To figure out the performance of SIMA-MPAS in reproducing the precipitation frequency distribution, we combine all the daily data from all the grid points at each coastal State (California, Oregon, and Washington) to calculate the frequency of daily precipitation by intensity (Figure 5). SIMA-MPAS captures a reasonable

distribution of precipitation intensity with respect to PRISM and Livneh observations, with smaller biases than WRF over California and Oregon regions, particularly at more extreme values (such as when daily intensity exceeding 20 mm/day). We also notice that over the Washington region, the biases for SIMA-MPAS and WRF are at similar magnitudes compared to the observations, although the two observations also show some uncertainties at the upper tail distributions.

Further, when examining the precipitation days with intensity less than 10 to 15 mm/day, SIMA-MPAS shows a close match to observations, while WRF tends to slightly underestimate the probability. For more extreme precipitation days, models tend to diverge in terms of the behaviors with SIMA-MPAS showing some underestimation over California and Washington regions (for average of ~14%, ~7% and ~18% bias for days when intensity exceeds 20 mm/day and less than 60 mm/day for California, Oregon, and Washington respectively). WRF generally overestimates the heavy precipitation frequency to a much larger extent (for an average bias of ~42%, ~51% and ~18% for California, Oregon, and Washington respectively). The sign of the biases is consistent with the previously discussed mean precipitation biases. It is not known to us why the biases in SIMA-MPAS are smaller than WRF. One hypothesis that would limit precipitation intensity is that SIMA-MPAS has strict conservation limits for energy and mass throughout the model, which are not present in WRF. This is a subject for future work, but may also be dependent on the specific WRF physics options used. We acknowledge that the initialization without nudging conditions in SIMA-MPAS simulations does not necessarily reproduce monthly or higher time variability but is able to get the seasonal means and distributions. We also acknowledge that the interannual variability and the sample size of the ARs could also affect the results of landfalling precipitation. Still, those analyses further testify the capability of using SIMA-MPAS for precipitation studies, giving us good confidence in using SIMA-MPAS for storm events studies.

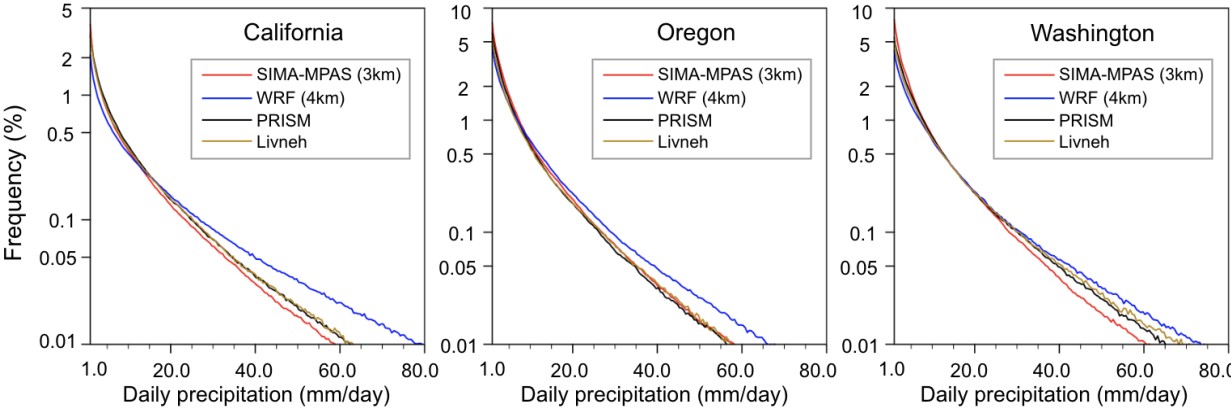

**Figure 5: Probability distribution of daily precipitation intensity.** All the daily datasets from the five wet seasons for all grid points in each State are used to construct the distribution statistics. The blue lines refer to WRF reference data, the black lines are for the PRISM observation, the dark golden line refers to the Livneh observation, and the SIMA-MPAS results are in red-colored lines. The SIMA-MPAS model data is regridded to the same resolution as the PRISM grid spacings (i.e.,

4 km). The x-axis starts from 1mm/day and the y-axis is transformed with a logarithmic scaling for better visualization of the upper tail distribution.

**3.2.2 MG2 vs. MG3 microphysics for simulated precipitation in SIMA-MPAS**

We would like to point out that we have used the default microphysics scheme-MG2 (Gettelman et al., 2015) when configuring those experiments from the CESM2 model. We acknowledge that MG3 (including rimed ice, graupel in this case) could be a better option with the rimed hydrometeors added (see Gettelman et al., 2019) especially when pushing to mesoscale simulations and for orographic precipitation. In detail, Gettelman et al 2019 found that the addition of rimed ice improved the simulation of precipitation in CESM at 14km resolution with wintertime orographic precipitation, due to altering the timing of precipitation by more correctly representing the pathways for precipitation formation with higher resolved scale vertical velocities. To fulfill this caveat but still make the best use of current simulation data, we have conducted another three experiments using the MG3 microphysics scheme for three wet seasons (1999-2002). Similar diagnostics have been performed as in the previous part but for the results from these three wet seasons (as shown in Figure 6).

Overall, the precipitation statistics are well represented in SIMA-MAPS compared to observations both with MG2 and MG3 when evaluating from the same three wet seasons. Although still outperforming WRF output, we do recognize that MG2 tends to underestimate heavy precipitation frequency in certain regions compared to observations, while MG3 produces more intense precipitation with some overestimations over heavy-precipitated regions, mostly over the Cascade Range and Coastal Range (Figure 6a). From the frequency distributions (Figure 6b), it can be seen that MG2 and MG3 microphysics both perform well over the study region. Specifically, MG3 produced stronger precipitation than the MG2 output over the Washington region showing a closer match to the observations than MG2 results. Due to interannual variability, we still need to investigate more different cases, and it is our next-step plan to further investigate the model performance with more testbeds.

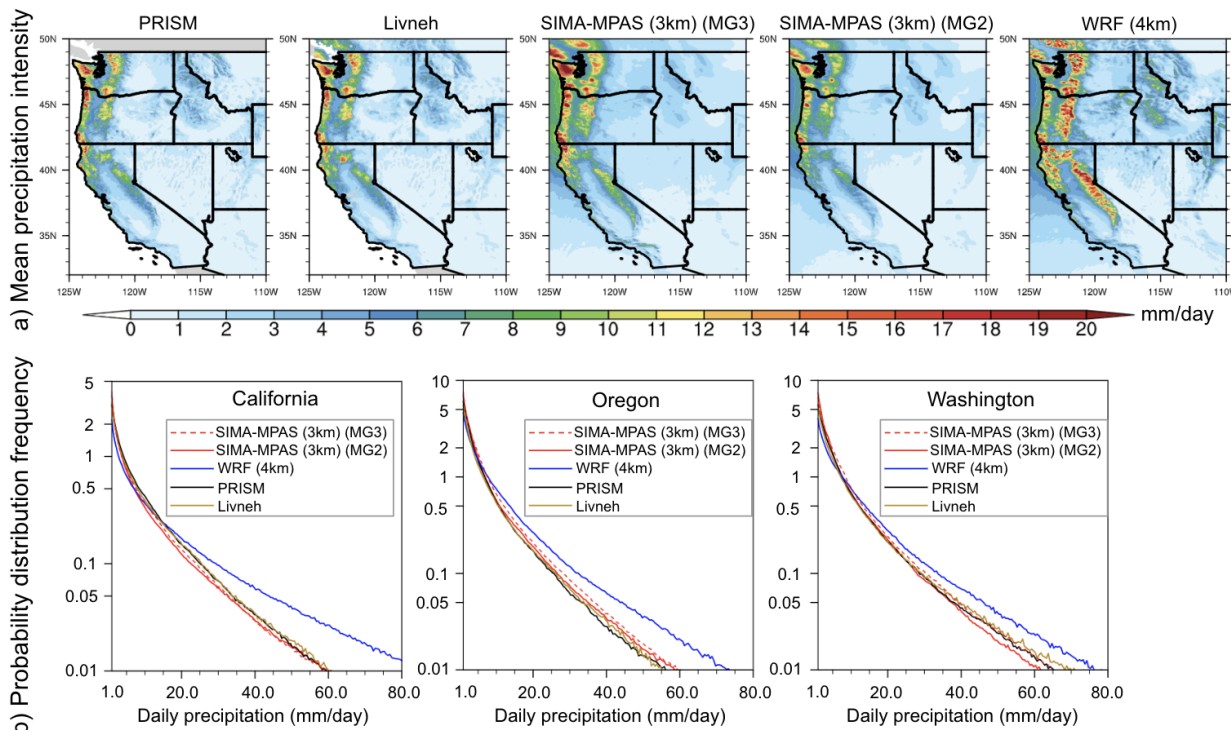

**Figure 6: MG2 vs. MG3 microphysics used in SIMA-MPAS for the wet-season precipitation over western US (1999-2002).** a) mean precipitation intensity; b) Probability distribution of daily precipitation frequency, like Figure 5 but for three wet seasons with SIMA-MPAS (MG3) added in dashed red lines; Again, the SIMA-MPAS model data is regridded to the same resolution as the PRISM grid spacings (i.e. 4 km).

## 3.3 Accumulated snowpack features

Snowpack characteristics have remained poorly represented in global climate models, lacking high-resolution terrain realization, fine-scale land-atmosphere coupled processes and interactions with snow's complicated thermal and hydrological properties (DeWalle & Rango 2008; Liu et al., 2017; Kapnick et al., 2018). Facing this long-standing issue, we expect that with much improved precipitation features, temperature, and substantially better-resolved complex terrains, snowpack features can be much better represented in CESM. Here, we have compared the accumulated snow water equivalent (SWE) results, which refer to the total accumulated snow from mid-Nov to mid-March (based on daily output), and then averaged over the five seasons (see Figure 7). By comparing with the gridded snow water equivalent observational data, it shows that SIMA-MPAS (MG2) can produce much improved estimation of the snowpack over the mountainous regions, with less overestimation than WRF simulations at similar resolution. However, the overestimation is notable for both SIMA-MPAS and WRF simulations, bringing the further need in investigating the land-air interactions in rain/snow processes and partitions from the precipitation contribution. In general, SIMA-MPAS can simulate reasonable spatial details for snowpack distribution over mountainous regions (mainly over the Cascade Range, Coastal Range, Sierra Nevada, and the

532 Rocky Mountains) with positive bias over the northern Cascade Range and certain Sierra Nevada
mountainous regions.

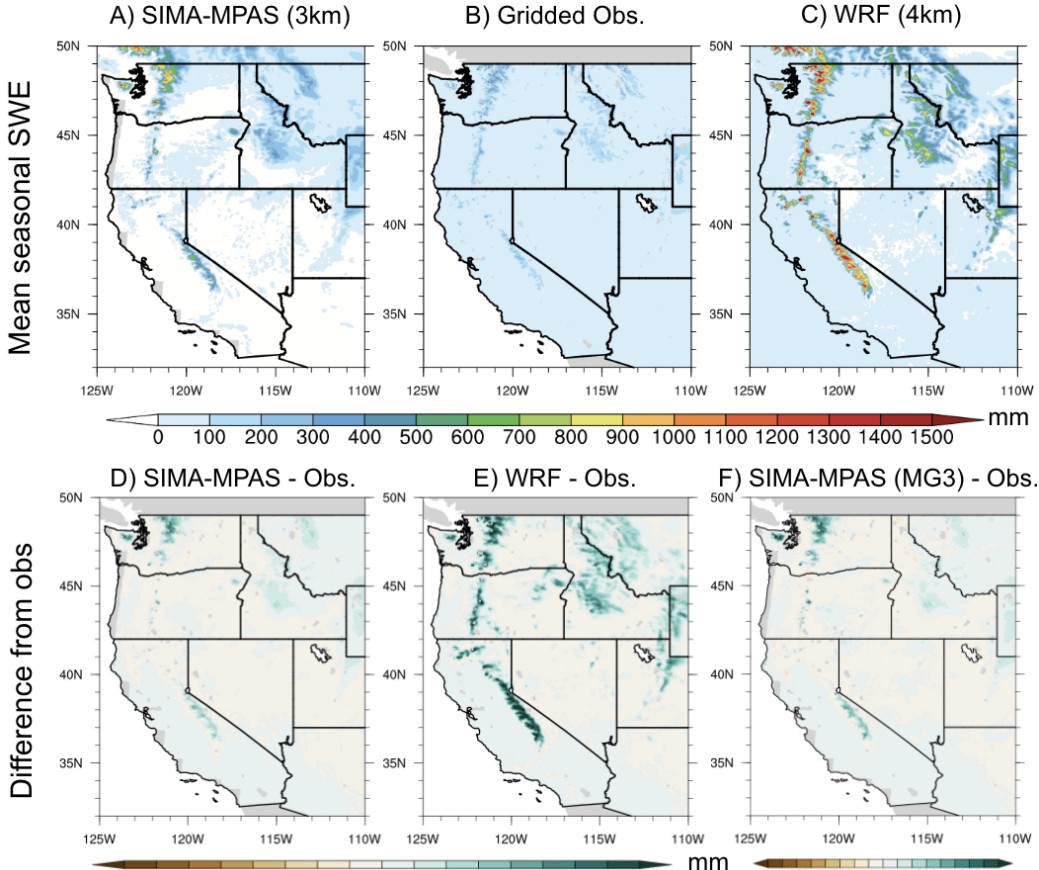

**Figure 7: Wet-season snow water equivalent (SWE) over western US.** First row: Seasonal
mean SWE averaged over (1999-2004) from A) SIMA-MPAS, B) Gridded observation for SWE
as described in the section 2.2, and C) WRF data; Second row (D, E, F): Absolute differences from
observation with all data regridded to 4 km for SIMA-MPAS and WRF averaged over (1999-
2004), and SIMA-MPAS (MG3) averaged over (1999-2002).

As the snowfall is dominated by the near-surface temperature and precipitation values, we have
examined the 2m temperature (T2) here to see how well temperature is captured in SIMA-MAPS.
In Figure 8, the mean T2 (T2mean) is shown averaged over all simulated wet seasons. In general,
near-surface temperature results from SIMA-MPAS are overall matched with observations across
varied climate zones including coastal areas, agriculture, desert regions, inland and mountainous.
However, we also notice that SIMA-MPAS tends to be warmer over most places (with the
averaged bias of about 0.65°C over the plotted domain), except over very high mountain top ranges
with cooler bias. On average, the difference for the regions with warmer biases is about 1.35°C
and the difference for those areas with cooler biases is about -0.99°C when compared to PRISM
data. On the contrary, WRF tends to be cooler in most regions except the southern part of Central

Valley and some desert regions in the southwest US (the average bias is about -1.84°C over the plotted domain). We have also investigated the T2 bias in the 120km simulations to see if this is a consistent model bias. By comparing FV and MPAS together (Figure S7), it turns out that SIMA-MPAS tends to be warmer with higher net surface shortwave and longwave fluxes over the wet-season period discussed here (Figure S8). Still, overall, the land model coupled with the atmosphere also does a good job here under a realistic topography.

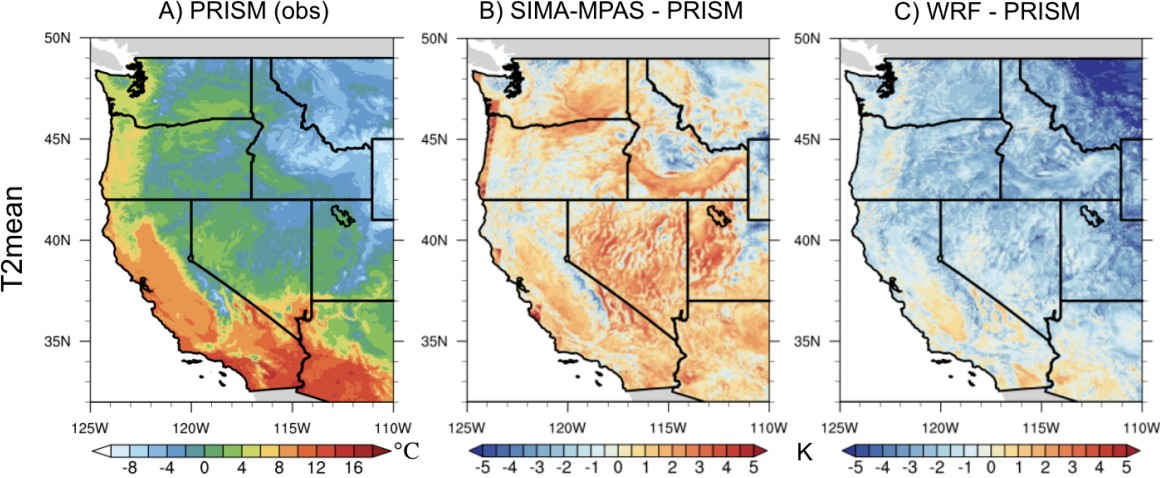

**Figure 8: Daily mean 2m air temperature (T2mean) averaged over (1999-2004, Nov-March).** A) PRISM observation dataset; B) and C) The differences between SIMA-MPAS and WRF from PRISM respectively; (Note: for difference plot, all data are regridded to the same resolution as PRISM).

**3.4 Large-scale moisture flux and dynamics**

Further, we have investigated the wind profile that directly connects to the subtropical to middle latitudes moisture fluxes over the northeast Pacific and the hitting western US regions. First, we have examined the cross sections of zonal and meridional wind patterns (at 130˚W, near the western US coast) at both 60-3km and 60km to determine the dynamic changes with the refinement mesh (Figure 9). As we can see, the mean westerly zonal winds are about 10% stronger at the jet stream level near 200-250hPa in 60-3km simulations compared to the 60km results. The mean meridional wind (dominantly southward) however is weaker in 60-3km simulations than the 60km ones. The precipitation over the western US coast is largely associated with the concentrated water vapor transport over the North Pacific, known mainly in the form of atmospheric rivers (Rutz et al., 2014). It is our further interest to investigate the wind dynamics transitioning from coarse-scale to mesoscale in future work. Another source of the precipitation uncertainty We would like to acknowledge the sensitivity from the physics timestep (see Figure S9) when comparing the precipitation in 60-3km simulations (a shorter physics time-step) to the 60km results at the regions with the same grid resolutions.

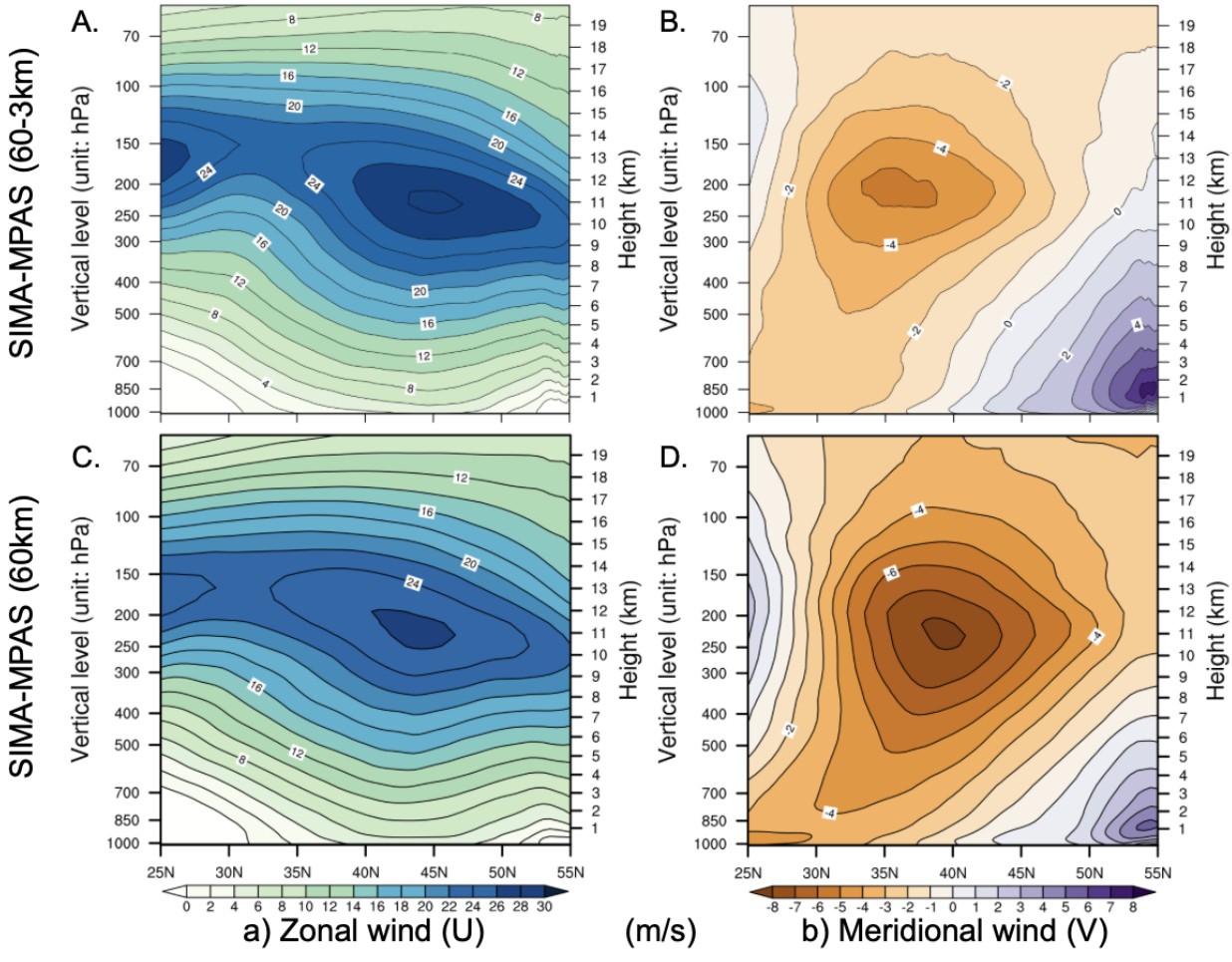

**Figure 9: Composite wind profile along western US coast (cross-section at 130W, near the western US coast) (averaged over 2000-2002, Nov-March).** a) Mean latitude-height cross-section of zonal winds (m/s) for SIMA-MPAS 60-3km (panel A) and 60km (panel C); b) similar as a), except for meridional winds (panel B and D).

In Figure 10, we further examine the large-scale moisture flux pattern from the integrated water vapor transport in the set of simulations with and without regional refinement. It can be seen that the spatial pattern of the moisture flux is generally similar between those two sets of experiments, dominated by the zonal winds (see Figure 9). If checking the IVT values along the longitude of 130°W, the differences (about 3% on average) are quite small along the WUS extent. With the large-scale dynamics and local fine-scale processes well integrated into this nonhydrostatic global climate model, it gives confidence in precipitation reproducing and predicting across the weather and climate scales.

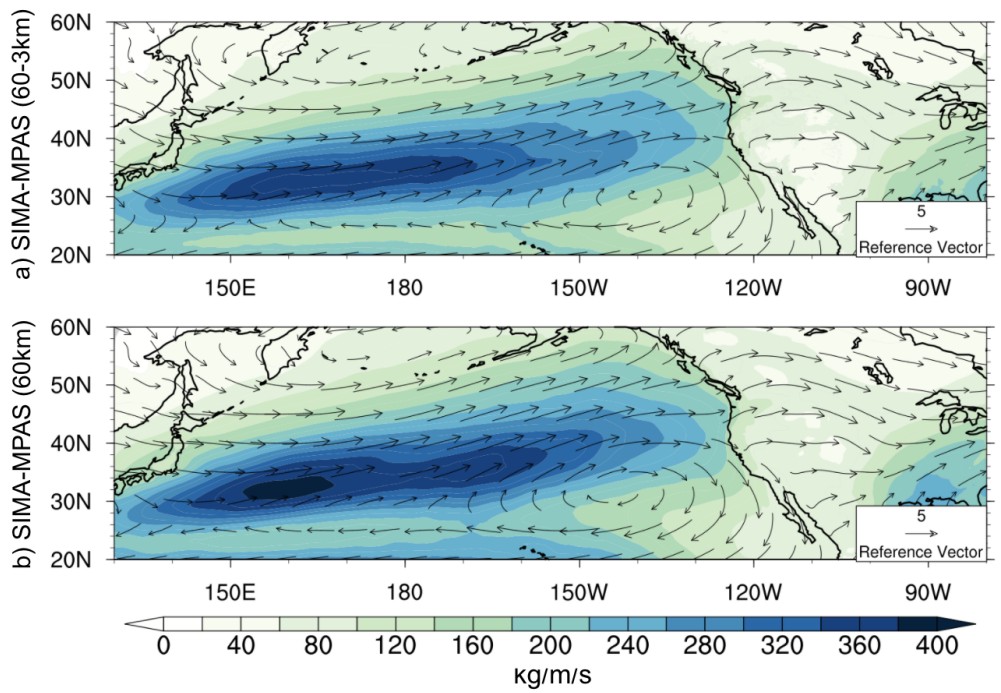

597

**Figure 10: Mean instantaneous vertically integrated water vapor flux transport over western US (2000-2002, Nov-March):** a) SIMA-MPAS 60-3km and b) SIMA-MPAS 60km. Wind is overlaid for the averaged lower levels (height from ~500m to ~2000m).

## 4 Summary and discussion

In this study, we describe SIMA-MPAS, which is built upon the open-source Community Earth System Model (CESM) with a nonhydrostatic dynamical core, the Model for Prediction Across Scales (MPAS), We would like to try to answer several questions about the performance of this new generation model when applying at convection-permitting resolutions and when bridging both weather and climate scale simulations in a single global model. We have chosen the western US as our study region to examine the precipitation features in SIMA-MPAS at fine scales and how the model performs when compared to both observations and a regional climate model.

To answer those questions, we have designed and conducted a set of experiments. First, we have tested CESM at the same coarse resolution using both MPAS as the nonhydrostatic core and finite-volume as the hydrostatic core for multiple years of climatology. Secondly, and, as the focus of this work, a variable resolution mesh is configured with 3km refinement centered over the western US. We have done five separate wet-season simulations to get the precipitation statistics. In addition, we have also included uniform 60km simulations from the model for two seasons.

We first evaluated the mean climate in SIMA-MPAS to see how that compares to the hydrostatic model counterpart (here, SIMA-FV). The diagnostics show that MPAS simulations have a very

similar climate to FV simulations. SIMA-MPAS has slight increases in cloud fraction and precipitation at the higher vertical resolution, while SIMA-FV has little change or slight decreases in cloud fraction. Overall, SIMA-MPAS produces a reasonable climate simulation, with biases relative to observations that are not that different from SIMA-FV simulations, despite limited adjustments being made to momentum forcing and no adjustment of the physics has been performed.

When compared to both observations and a traditional regional climate model at similar fine resolutions for mean and heavy precipitation behaviors, SIMA-MPAS can capture the spatial pattern and mean intensity (with the spatial correlation of about 0.93 relative to PRISM), which is also comparable to WRF results. We do notice there are some underestimations mostly in SIMA-MPAS and overestimations mostly in WRF. Further, SIMA-MPAS captures the distribution of precipitation intensity with respect to observations with smaller biases than WRF over California and Oregon regions, particularly at more extreme values. With additional experiments, SIMA-MPAS with MG3 microphysics (graupel) produces stronger precipitation than the MG2 version (as used in other experiments in this study as the default microphysics scheme) and the MG3 results also well presented the precipitation statistics for both spatial mean and frequency distribution. The difference between MG3 and MG2 is the rimed hydrometeors added to the MG3 (see Gettelman et al., 2019 for detailed descriptions), which could matter more when pushing to mesoscale simulations and for orographic precipitation. We also acknowledge the interannual variability and it is our next-step plan to further investigate the model performance with more testbeds.

We further show that SIMA-MPAS can produce much improved estimation of the snowpack over the mountainous regions compared to coarse resolutions, with less overestimation than WRF simulations at similar resolution. In general, SIMA-MPAS can simulate some reasonable spatial details for snowpack distribution over mountainous regions (mainly over the Cascade Range, Coastal Range, Sierra Nevada, and the Rocky Mountains) with positive bias over the northern Cascade Range and certain Sierra Nevada mountainous regions. The overestimation is notable for both SIMA-MPAS and WRF simulations, needing further investigations. We also notice that SIMA-MPAS tends to be warmer over most places, except over very high mountain top ranges with cooler bias.

The results further testify the capability of using SIMA-MPAS for precipitation studies, giving us good confidence in using SIMA-MPAS for storm events studies. We focus on multiple-season statistics for model performance. Given the large-scale dynamics and local fine-scale processes well integrated into this nonhydrostatic global climate model, it shows promise in precipitation reproducing and predicting across the weather and climate scales. It is our further interest to investigate the wind dynamics transitioning from coarse-scale to mesoscale in future work and to

further investigate the model performance with more testbeds for convection-permitting weather and climate systems across scales.

**Data and code availability:** The data and codes used in this work are available for access from this DOI link: https://doi.org/10.5281/zenodo.6558578. The model used in this study can be downloaded from the open-shared link: https://github.com/ESCOMP/CAM.

**Author Contributions:** XH and AG designed the study and the experiments. All authors contributed to the work in the model development. XH performed the simulations with assistance from AG, MC, WS, PL, and AH. XH and AG contributed to the investigation and visualization. XH prepared the manuscript with review and edits from AG, WS, PL, and AH.

## Acknowledgments

We thank the editor and two anonymous reviewers for their comprehensive comments that helped to improve the quality and presentation of this manuscript. We acknowledge the open-shared dataset used in this study including CERES EBAF products (https://ceres.larc.nasa.gov/data/), GHCN Gridded V2 CPC data provided by the NOAA/OAR/ESRL PSL (https://www.psl.noaa.gov/data/gridded/data.ghcncams.html), PRISM (https://prism.oregonstate.edu/) and Livneh (http://cirrus.ucsd.edu/~pierce/nonsplit_precip/) observations, and WRF simulations at 4km (https://rda.ucar.edu/datasets/ds612.5/). We acknowledge the funding support from NSF funded project Earthworks (award number is NSF 2004973). We also acknowledge the partial support from the National Center for Atmospheric Research (NCAR), which is a major facility sponsored by the NSF under Cooperative Agreement 1852977, and the high-performance computing support and data storage resources from the Cheyenne supercomputer (doi:10.5065/D6RX99HX) provided by the Computational and Information Systems Laboratory (CISL) at NCAR.

**Competing Interest Statement:** The authors have no competing interests to declare.

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
