# Peer review of "Advancing Precipitation Prediction Using a New Generation Storm-resolving Model Framework - SIMA-MPAS (V1.0): a Case Study over the Western United States"

_Geoscientific Model Development, 2022_

## Author Comment (AC1)

**Reviewer 1:** This paper presents some case studies using the SIMA-MPAS GCM, which is a combination of the atmosphere and land component of the Community Earth System Model (CESM) and the nonhydrostatic MPAS dynamical core using the SIMA framework. The main simulations are performed at variable resolution with a 3km grid spacing covering the western US and a 60km grid spacing for the remaining globe. When compared to observations, SIMA-MPAS shows more realistic precipitation intensities, snowpack cover, and a smaller 2m temperature bias than simulations with the regional climate model WRF at a similar resolution (4km grid spacing).

The study provides confidence in the ability of SIMA-MPAS to produce realistic global climate simulations with variable grid spacing and the use of storm-resolving scales in regions of interest. This is a nice achievement considering the non-trivial coupling between the CAM physics package and the MPAS dynamical core (as described in the paper) and it encourages further research using this model framework. Therefore, I think the paper would be a useful contribution for the atmospheric modeling community.

We very much appreciate the reviewer's helpful input and the positive feedback. We have made substantial changes to the manuscript as enumerated below (blue colored). We have completed substantial revisions in the form of text revisions, figure modifications, additional simulations and analysis, to address each individual comment directly. We think the manuscript has been improved in many aspects.

However, there are some aspects of the paper that should be improved before publication.

Major points:

- A significant part of this work is the comparison of the model output with observational data, which is used as justification for the fitness of the model. However, the description of the different observational datasets is very brief (or non-existent). Without going through the references and performing their own literature research, many questions are unanswered to the audience. What are the respective resolutions of the datasets? How is the data obtained (i.e., what kind of product is it)? Are there any known biases (here, one gets the impression that they are the ground truth)? How do they compare to other observational products? Since the observations play such a significant role in this paper, I would expect a more detailed description of the respective datasets.

  Thank you for the suggestion. We have added detailed descriptions of the observational datasets (one separate paragraph for each data product) in Section 2.2. As suggested by the second reviewer, we have also added the recently released Livneh precipitation data (Pierce et al., 2021) as another gridded observationally-based precipitation dataset to better account for extreme precipitation. The added text is also copied below:

  "Detailed descriptions of the open-shared datasets used in this study are given below:

- CERES EBAF data products: we use gridded data from the Energy Balance And Filled (EBAF) product from the NASA Clouds in the Earth's Radiant Energy System (CERES), described by Loeb et al (2018). CERES provides high quality top of the atmosphere radiative fluxes and cloud radiative effects, as well as consistent ancillary products for Liquid Water Path (LWP) and cloud fraction. We start with monthly mean gridded products at 1˚ and make a 20 year climatology from 2000-2020.

- GHCN_CAMS Gridded 2m air land temperature: global analysis monthly data from NOAA PSL comes with resolution at 0.5 x 0.5°. It combines two large networks of station observations including the GHCN (Global Historical Climatology Network version 2) and the CAMS (Climate Anomaly Monitoring System), together with some unique interpolation methods (https://psl.noaa.gov; Fan and Van, 2008).

- PRISM observed data: the Parameter-elevation Regressions on Independent Slopes Model (PRISM) gridded observed data for daily precipitation and daily 2m air temperature is used at 4 km grid resolution (Daly et al., 2017; https://prism.oregonstate.edu/). Covering Continental U.S., PRISM takes the station observations from the Global Historical Climatology Network Daily (GHCND) data set (Menne et al., 2012) and applies a weighted regression scheme that accounts for multiple factors affecting the local climatology (Daly et al., 2017).

- Livneh gridded observationally-based precipitation dataset: in addition to PRISM data, to better account for extreme precipitation, a recently released Livneh precipitation data (Pierce et al., 2021; http://cirrus.ucsd.edu/~pierce/nonsplit_precip/) is also used for model evaluation. The data (~6km grid resolution) is shown to perform significantly better in reproducing extreme precipitation metrics (Pierce et al., 2021).

- Snow water equivalent (SWE) data over the CONUS: this is the observational data product we use for snowpack diagnostics. The data is available from National Snow and Ice Data Center (NSIDC) (at https://nsidc.org/data/nsidc-0719/versions/1). The product provides daily 4km SWE from 1981 to 2021, developed at the University of Arizona. The data assimilated in-situ snow measurements from the SNOTEL network and the COOP network with modeled, gridded temperature and precipitation data from PRISM (Zeng et al., 2018; Broxton et al., 2019).

- CONUS (Continental U.S.) II high resolution climate simulations: The WRF (Weather Research and Forecasting) nonhydrostatic model simulations we used for comparison are from Rasmussen et al. (2021) (accessible at https://rda.ucar.edu/datasets/ds612.5). Its horizontal grid resolution is 4 km with forcing from the mean of the CMIP5 model for both present (1996-2015) and future (2080-2099) mean climate, with hourly output. For the study region we focus on here (i.e. over the western US), the simulations provide a more realistic depiction of the mesoscale terrain features, critical to the successful simulation of mountainous precipitation (Rasmussen et al., 2021)."

- I find Fig. 3 confusing. For E) "Land 2m Temperature", the legend says that EBAF 4.1 was used. However, in Sect. 2.2 you state that CERES EBAF was only used for cloud and radiation fluxes properties, whereas GHCN (which is mentioned nowhere except in Sect. 2.2) and/or PRISM was used for 2m temperature. So either the legend in the figure is wrong or the description in Sect. 2.2 is wrong. Please adjust and clarify. Also, if you have 2m temperature from PRISM and GHCN, why would you not include both into the analysis? Furthermore, I would suggest using a clearly different color for the observations to make it stand out more.

Thank you for catching this. We have revised the caption for Figure 3 clarifying that the observations for the land 2m air temperature is from GHCN_CAMS Gridded data. PRISM observation is not included here as it only covers the Continental U.S. region. For clarification, we have added the relevant data information in detail in Section 2.2. As suggested, we have updated Fig. 3 using a clearly different color (in green) for the observations (revised figure copied below).

[Figure]

- I'm missing any information on the performed regridding for Fig. 5 and Fig. 6. Has the model and observational data been regridded for the analysis? If yes, to what grid and how? If no, how do you account for different grid spacings? This information is crucial for such an analysis (and the reproduction of the results!) and it should be clearly mentioned in the paper.

Thank you for pointing this out. Yes, we have regridded the model data to the same grid resolution as the PRISM observation (i.e. 4 km). For the regridd method and procedure, we first regridded CAM-MPAS data from unstructured grids to regular rectilinear lat/lon grids at 0.03 degree with ESMF software functions, and then regridded to the same grid spacings as the PRISM using the bilinear interpolation with relevant CDO command.

We have now added this information to Section 2.2: "For the regridd method and procedure, first CAM-MPAS data is regridded from unstructured grids to regular rectilinear lat/lon grids at 0.03 degree, and then the rectilinear data is regridded to the same grid spacings as the PRISM using the bilinear interpolation." and mentioned that "The SIMA-MPAS model data is regridded to the same resolution as the PRISM grid spacings (i.e. 4 km)." when it applies.

- SIMAS-MPAS (3km) with MG2 microphysics seems to perform very well in spatial representation of precipitation (Fig. 4) and daily precipitation frequency (Fig. 5) when compared to observations over 5 seasons. However, when only looking at one season for the comparison with MG3, the MG2 version underestimates heavy precipitation frequency. So there seems to be quite a bit of variability and this does not exactly provide confidence in the robustness of the results, especially for Fig. 6. Is one season really enough to conclude that MG3 performs better? Maybe MG3 would overestimate heavy precipitation frequency over all 5 seasons? You also state in the paper that this issue requires more investigation. Therefore, I'm not sure whether it's wise to include these results that prominently in the paper. Maybe these results would be better suited for the appendix.

Thank you for the suggestion. We agree that the variability for using one season in assessing MG3 performance. To better understand this, we have added two new simulations as we can (as each simulation is computation intensive) for another two wet seasons using MG3 microphysics. We have updated the results section 3.2.2, combining the three seasons in total for MG3. Overall, the precipitation statistics are well represented in SIMA-MAPS compared to observations both with MG2 and MG3. We do recognize that MG2 tends to underestimate heavy precipitation frequency in certain regions compared to observations, while MG3 shows a closer match in those cases with more intense precipitation produced. Changes are made to the manuscript. The updated Fig. 6 is copied below.

[Figure]

(updated) Figure 6: MG2 vs. MG3 microphysics used in SIMA-MPAS for the wet-season (Nov-March) precipitation over western US (1999-2002).

- Is the SST and ice sheet for Set A constant? Or are the forcings from different years or only from year 2000? How is the model initialized? From the paper it is not entirely clear to me how the mean climatology is obtained. Also, the last sentence at Section 3.1 does not make much sense to me (lines 224-225). Please clarify.

The SST and ice data for Set A are prescribed at the same yearly climatology (i.e. 12 months) with mean from the time period 1995-2005. We have clarified the referred sentence in Section 3.1 to "Simulations results are averaged over the five years output under the present day climatology (with SST and ice forcings from the mean of time period 1996-2005)".

Minor points / typos:

- Lines 14-16 in the abstract read like SIMA is the atmospheric component of CESM, whereas from reading the introduction and the website, SIMA is just a framework that allows for the coupling of different components. Please clarify or reformulate.

Thank you for catching this. We have rephrased the referred sentence to "This study uses a state-of-art storm-resolving GCM with a non-hydrostatic dynamical core - the Model for Prediction Across Scales (MPAS), incorporated in the atmospheric component (Community Atmosphere Model, CAM) of the open-source Community Earth System Model (CESM), within the System for Integrated Modeling of the Atmosphere (SIMA) framework."

- Section 3.4 & Fig. 9: I would at least expect a sentence about the use of gravity wave drag parameterization for the different simulations (I assume the 60km uses one, whereas it's not really necessary for 3km), as this will likely have an effect on the strength of the jet.

Thank you for the suggestions. We note in the discussion of the simulations in section 2.2 under topography how the gravity wave drag scheme operates: The orographic gravity wave drag scheme in SIMA-MPAS (used in CESM2-CAM6) uses a 'sub-grid' orography to force the scheme. The 'sub-grid' orography is calculated for each grid cell from a standard high resolution (1km) Digital Elevation Model and used to drive the scheme. Thus the sub-grid orography forcing is small at 3km, and is larger at 60km. The theory being that more of the drag and waves are resolved at higher resolution. So the overall drag should be somewhat similar with scale, but partitioned differently between resolved and unresolved. We have added those explanations to the main text.

- I would reverse the color bar for Fig. 4 b) and c). In Fig. 4 a), red means more precipitation and blue means less. For the differences it is the other way around. I believe it would make the plots easier to read to reverse it for b) and c) (as you have done it in Fig. 8).

Thank you for the suggestion. We have updated Fig. 4 with reversed colorbar in b) and c). The revised Fig. 4 is copied below.

[Figure]

- The terms "non-hydrostatic" and "nonhydrostatic" are both used in this paper.

For consistency, we have replaced "non-hydrostatic" with "nonhydrostatic" throughout the text.

- The term SST is used without definition.

Thank you. We have added this information.

- Lines 122, 143, 149, 181, ….: "We would" instead of "We'd"

Thank you. We have corrected those wordings.

- Line 421: "vertical wind patterns" sounds like you have analyzed vertical winds. Maybe use "cross sections of zonal and meridional winds" or something similar.

Thank you for pointing out this. We have rephrased this accordingly.

---

## Author Comment (AC2)

**Reviewer 2:** Huang et al. in "Advancing Precipitation Prediction Using a New Generation Storm-resolving Model Framework - SIMA-MPAS (V1.0) a Case Study over the Western United States" evaluates several versions of the SIMA-MPAS model (e.g., vertical resolution, horizontal resolution, and microphysics schemes) that has been recently implemented into a widely used Earth system model, CESM2. The authors evaluate the five-year, near-term historical model simulations, mostly, over the western United States (save for a global comparison to the "work horse" dycore in CESM2-CAM6, CAM-FV) and compare model performance to a traditional regional climate model and observationally based gridded products.

Overall, I think the paper fits within the scope of GMD and could be, given more work, a valuable contribution to the scientific community. The model advancements/developments (as described in lines 121-141), experimental designs, and data production aspects of this manuscript are clear and robust (save for questions about WUS interannual variability). However, the messaging in the manuscript was very choppy and appeared hastily put together. I have tried to provide constructive feedback to overcome this but highly encourage the entire authorship team to provide a more thorough internal edit during the next revision. As a result, I think there are several major revisions that need to happen prior to this paper being accepted.

We very much appreciate the positive feedback, comprehensive comments and suggestions from the reviewer. We have comprehensively addressed the respective concerns and comments via a combination of text changes, figure updates, and new/additional analyses. These revisions are enumerated below (blue colored). We think the manuscript has been strengthened in many aspects.

Major and Minor Revisions

Line 12-13 – Delete "…for predictions to be useful…" You might also want to bring up the scale-awareness aspects of parameterizations that shape precipitation intensity, duration, and frequency.

Thank you. We updated this sentence accordingly.

Line 17 – Delete "For mean climatology" and add "…with reasonable energy and mass conservation on climatological timescales"

Thank you. We have updated this accordingly.

Line 19 – "We mainly investigate…" this was an awkward sentence transition. I think what you're trying to say is that you wanted to prove comparable energy/mass balance performance between CAM-FV (workhorse dycore) and SIMA-MPAS (newly developed dycore) and that this coarse resolution performance should give confidence to the community that SIMA-MPAS can then be evaluated at finer resolution?

We have clarified this. It now reads: "With the comparable energy and mass balance performance between CAM-FV (workhorse dycore) and SIMA-MPAS (newly developed dycore), it gives

confidence in SIMA-MPAS's applications at a finer resolution. To evaluate this, we focus on how the SIMA-MPAS model performs when reaching storm-resolving scale at 3km."

Line 21 – "Effectively" or "Efficiently"?

Thank you. Rephrased to "Efficiently".

Line 22 – Change to "…60 km over the rest of the globe…"

Thank you. Updated.

Line 23 – Delete "precipitation details"

We have deleted "details".

Line 24 – What are "temperature features"?

We have deleted "features" and changed the words to "near-surface temperature".

Line 26-27 – "We compared and evaluated …" this whole sentence should be presented earlier, and more specificity is needed.  What observations?  What traditional model?

Thank you for the suggestion. We have moved this sentence earlier and added more specific information. The updated sentence reads "We evaluated the model performance using satellite and station-based gridded observations with comparison to a traditional regional climate model (WRF, the Weather Research and Forecasting model)."

Line 29-30 – "The" or "This"?

Corrected to "This".

Line 44-45 – This was another awkward sentence transition.  I would delete "Given the recent development of Earth system model frameworks…" and start with "Advances in computer power have now enabled climate models to be run with non-hydrostatic dynamical cores at "storm-resolving" scales, on the order of a few kilometers…"

Thank you and we agree in this way it reads much better. The sentence has been rephrased accordingly.

Line 51 – What are the differences between "structures" and "platforms"?

We have removed the word "platforms" for unnecessary confusion.

Line 54 – Delete "one of the leading Earth system models" (this point is proven by the next part of the sentence)

Thank you. Deleted.

Line 57 – CESM has been used in many other regional climate modeling contexts too. Please thoroughly cite other uniform high-resolution and variable-resolution CESM-SE (spectral element core) studies performed over the years too.

We have added some other studies for a general idea of the literature without an overwhelming list. Cited as "(such as Small et al., 2014; Zarzycki et al., 2014, 2015; Rhoades et al., 2016; Huang et al., 2016, 2017; Bacmeister et al., 2018; Gettelman et al., 2018, 2019; Van et al., 2019)"

Line 58 – Change "modeling" to "model"

Corrected.

Line 58 – Either use "more recently" or "over the past decade"

Thank you. Deleted "over the past decade".

Line 64 – If the authors state "number of studies" I think more than one study should be cited.

Thanks. We have added a few more studies, cited as "(e.g., Rauscher et al., 2013; Rauscher & Ringler, 2014; Sakaguchi et al., 2016)".

Line 92-105 – Can the authors, in a paragraph or so, provide hypotheses on how any portions of the new modeling system where there is/isn't scale awareness, particularly when transitioning from ~120km to ~60km to ~3km, could impact regional precipitation statistics (e.g., intensity, duration, frequency, phase, extremes, etc.), with a particular focus over the WUS? Also, would there be any resolution dependent land surface related feedbacks to the atmosphere that might arise?

These are good questions. We have added the following paragraph to the Methods Section 2.1, which reads:

"In terms of scale awareness, there are two aspects related to the model physics in the configuration that must be considered when employing regionally-refined meshes. First, features resolvable in the finer regions of the mesh may not be resolvable in the coarser regions of the mesh. These features, e.g. deep convection in this study, need to be parameterized in the coarse mesh regions and not parameterized in the fine mesh regions, typically with the parameterization reducing its adjustment gradually in the mesh transition regions. Second, the timestep used for the physics is the same over the entire mesh. i.e. in both coarse and fine regions, and the timestep in CESM-MPAS is chosen to be appropriate for the smallest grid, as indicated in Table 1. Within our simulations, the balance of deep convective (diagnostic) and stratiform (large-scale) precipitation changes with the mesh spacing. In addition, since the deep convective parameterization in CESM-MPAS has a closure with a fixed timescale, the parameterized convection produces less condensation in the coarse mesh regions compared to simulations with a larger timestep appropriate for the coarser mesh (Gettelman et al 2019). But in the simulations herein, most of the precipitation is strongly forced by the large-scale flow, with the larger condensation hypothesized to lead to larger rain rates. This is particularly important over the WUS complex terrains. The large scale condensation scheme, part of the unified turbulence scheme (CLUBB: Golaz and Larson 2003) has internal length scales that

should adjust its distributions as the scale changes (less variance in the PDFs). Land surface related feedback is also resolution dependent with scale-aware surface heterogeneity and coupled land-atmosphere interactions to affect the phase and hydrological impacts resulting from the regional precipitation statistics."

Line 101 – "Predictions"

Corrected.

Line 103-104 – Change "snowpack statistics features" to "precipitation and snowpack statistics"

Corrected.

Line 114 – Observed SST and sea ice?

Thank you. Clarified to "prescribed observation-based SST (sea surface temperature) and ice."

Line 144 – The authors alternate between the use of "storm-resolving" and "convection-permitting", why?

Oh, no particular reason. These two terms are alternatively used to mean the same.

Line 145-147 - I think the authors should provide a justification (a few sentences) for why the WUS is chosen and not some other region of the world.

We have added such information and the updated sentence now reads "We have chosen the WUS (due to its hydroclimate vulnerability and complexity, heavily impacted by precipitation variability) as our study region to examine the precipitation features in SIMA-MPAS at fine scales during wet seasons."

Line 148 – Change to "…when compared with a traditional regional climate model against best-available observations and observationally-based gridded products at similar…"

Thank you and updated. That reads much better.

Line 149 – Delete "behavior". Also "heavy" or "extreme"?

Thank you. Deleted and changed "heavy" to "extreme" as a more proper word.

Set B, C, and D – I know model simulations can be quite costly, but given the large interannual variability of precipitation in the WUS, are five simulated years enough? Were 1999-2004 consistent with a near-term climatology of the region? If not, could the authors, at least, mention this somewhere as a potential constraint.

Good point. We have mentioned it in Section 2.1 (Methods and experiments) saying that "Given the interannual variability of precipitation over the WUS study region, we acknowledge that it is not our goal to reproduce the recent historical climatology but to evaluate the overall model performance."

Line 163 – Change "western us" to "WUS"

Updated.

Line 164 and 169 – Was there a reason that the authors did not want to assess summer/fall precipitation (e.g., convective storms, Monsoons, etc.), particularly over the Rocky Mountains? This may be addressed earlier on in the manuscript with an additional paragraph that describes hypotheses of model performance across resolution (or, simply, long-standing biases for particular seasonal precipitation, regardless of resolution).

Thank you for the comment. Although we did not focus on convective storms or summer/fall precipitation, other colleagues have been working on the convective storm cases as separate work (Relevant online abstract goes here?). We also introduce the questions/hypotheses we'd like to answer in the previous paragraph, which reads "..., we would like to figure out whether a nonhydrostatic dycore coupled global climate model can reproduce observed wet season precipitation over targeted refinement regions with heavy impacts. And will this new development and modeling framework perform better or worse than a mesoscale model at similar resolution? Those are important questions to answer given the long-standing biases in traditional hydrostatic GCMs for simulating heavy precipitation and extremes."

Line 171-174 – I'd suggest the authors provide more context as to why the addition of graupel could be an important driver of WUS precipitation statistics and should be evaluated as a unique, standalone, and resource intensive simulation (like Set A-C)

We have provided this context. The sentence reads "Specifically, Gettelman et al 2019 (i.e. the MG3 paper) show that even at 14km scale the inclusion of rimed ice changes the timing and location of precipitation in the Western United States due to the different fall speeds and lifetimes of graupel, which is formed when higher vertical velocities result. This effect is expected to be larger at 3km."

Line 166 – Delete "reanalysis" (redundant)

Thank you. Deleted.

Line 177 – Delete "(i.e., seconds)"

Deleted.

Line 178-179 – Could the authors provide a bit more justification for the physics timestep choice (or point to other studies)?

The physics timestep is 1800s for the ~1deg/120 km resolution experiments, and this is the default for CAM6 physics for the nominally 1 degree MPAS mesh. We reduce it to 900s for the 60km grid-space experiments, scaling it with reduced mesh spacing. Instead of using a 20 s timestep for the 60-3 km mesh as scaling would imply, we use a 120s physics timestep for the 60-3km experiments, in part to reduce computational cost and because other studies have shown acceptable results with this physics timestep at comparable mesh spacing (e.g. Zeman et al 2021). We also

recognize that the WUS precipitation as the focus of our study is predominantly orographically forced, whereas the physics-timestep-critical processes are related to unstable deep convection, perhaps lending support for a longer physics timestep in this application. We acknowledge the possible sensitivity of our results to the physics timestep and we will be examining this more in future work. The main text has been updated accordingly.

Line 180-181 – Which simulation cost this much? I imagine each simulation was different in cost. Maybe place this information in Table 1. Which high performance computing system? What was the simulation throughput per node? How large were the output data volumes? (I think this could be useful information to those folks interested in running these sorts of experiments on university clusters or … in the future when the model configurations are publicly available)

We have added the information that it is the 60-3km simulations' computation cost. We use the Cheyenne supercomputer for the simulations (see details in the Acknowledgments part) and each node has 36 cores. Each simulation costs differently due to many factors of course, such as the number of grid columns, the number of vertical levels, the timesteps, and different model dynamics and physics. We have now added the number of grid columns in the table for a better idea of the computation cost. The output data file size depends on how many 2D and 3D variables are saved and the output frequency. In this aspect, the number of the grid columns and vertical levels information in table 1 should be adequate for future use reference.

Table 1 caption is a bit colloquial.

Revised to "A list of experiments in this study and the key configuration information"

Figure 1 – I think the authors should be more specific in the caption, "SIMA-MPAS mesh configuration for the 60-3km experiments", to make the figure standalone.

Thank you for the helpful suggestion. Updated accordingly.

Section 2.2 – "Other datasets" is a bit broad. How about "Observations and observationally-based gridded products used to evaluate model performance"

Thank you. That reads very well. Updated.

Line 199-200 – PRISM and Daymet are "observationally-based". Also there are known issues with PRISM's ability to represent extreme precipitation events (which might be important when comparing a smaller subset of years) - https://doi.org/10.1175/JHM-D-15-0019.1 - "...In general, over the entire period, the gridded datasets performed reasonably well, with over 50% of median errors on individual days falling between −37% and 44% and water-year total errors within ±10% (Table 2). However, errors in individual storm events sometimes exceeded 50% for the median difference across all stations, and in some years, these underpredicted storms led to 20% error in water-year total median statewide snowfall (e.g., water-year 2008, Fig. 9)…" A path forward may be to use a recently released dataset that better accounts for extreme precipitation - https://doi.org/10.1175/JHM-D-20-0212.1 ( http://cirrus.ucsd.edu/~pierce/nonsplit_precip/ )

Thank you for the information and suggestions. We have updated the relevant figures adding the suggested recently released gridded observationally-based precipitation dataset as Livneh data. We also copied the updated figures below. Although the differences between PRISM and Livneh on average is small, it provides a more robust evaluation for both mean and extreme precipitation by having those two observational products. We have added additional discussions in the text for the updated figures and the new observational data has been described in the Section 2.2 in detail.

Figure 4a: Mean precipitation.

[Figure]

Figure 5: Probability distribution of daily precipitation intensity.

[Figure]

Line 199-200 - Daymet is not usually used for snowpack evaluation in the WUS. From what I've gathered, Daymet treats snowpack as more of an "energy budget fixer" term in the Daymet dataset generation process (discontinuities can be seen between Dec 31 and Jan 1). UofA ( https://nsidc.org/data/nsidc-0719/versions/1 ) and the UCLA ( https://nsidc.org/data/WUS_UCLA_SR/versions/1 ) datasets are more widely trusted/used and I'd encourage the authors to consider one (or both) of them for this evaluation.

Thank you for the information and suggestions. Given the discontinuities consideration in Daymet, we have replaced Daymet with one of the suggested SWE data provided by UofA as the observational data. We have updated the relevant figure and discussions with this SWE observation data replacement. This data has also been described in Section 2.2 in detail. We also copied the updated figure below.

Figure 7: Accumulated snowpack.

[Figure]

Line 196-209 – did the authors remap any of the datasets before comparison? Could the authors provide explicit details on the type of remapping and which "direction" was used (i.e., remap observations to models or models to observations or all datasets to a common resolution or …)

Thank you for pointing this out and a similar comment has been brought up by the other reviewer too. Yes, we have regridded the model data to the same grid resolution as the PRISM observation (i.e. 4 km). For the regridd method and procedure, we first regridded CAM-MPAS data from unstructured grids to regular rectilinear lat/lon grids at 0.03 degree, and then regridded to the same grid spacings as the PRISM. We have now added this information to this section (i.e. 2.2), which reads "For the regridd method and procedure, first CAM-MPAS data is regridded from unstructured grids to regular rectilinear lat/lon grids at 0.03 degree, and then the rectilinear data is regridded to the same grid spacings as the PRISM using the bilinear interpolation."

Line 205 – Delete "as a"

Thank you. Deleted.

Line 224-25 – This sentence is very confusing.

Simulations results are averaged over 5 years. Note that the simulations are forced with the same climatological monthly mean boundary conditions for sea surface temperature and greenhouse gasses every year to reduce interannual variability. We have clarified this in the text.

Figure 3 – is the caption correct (e.g., CESM2-CAM6)?  Is CAM6 used as an umbrella term for both SIMA-MPAS and CAM-FV?

We have updated this part of the caption, which now reads: "Zonal mean climatology from 5-year simulations with CESM2 and CAM6 physics using different dynamical cores and vertical levels."

Line 238-248 – Can the authors discuss why the largest differences (e.g., ice water path and cloud fraction) between CAM-FV and SIMA-MPAS occur in the midlatitudes (e.g., WUS)?

For clarification, we have updated Figure 3 and added two new supplemental figures (copied below). Overall, climate impacts come likely from the differences in near-surface wind, turbulent interactions near the boundary layer, and changes in temperature associated with the winds (in geostrophic balance) and to some extent due to different treatments of the surface topography. And none of the changes are outside the bounds of typical uncertainties in model physical processes (e.g. tuning). We have added the paragraph to the main text:

"When examining the spatial differences (Figure S2 and Figure S3), we further found that the differences in the wind over the oceans drive differences in aerosols (sea salt) which alter the aerosol optical depth and droplet concentration. The radiative effects come as a result of cloud fraction changes: high clouds and specifically ice water path for the longwave, low cloud and liquid Water Path for the shortwave. The signal in clouds is stronger at L32 (Figure 3, Figure S2), again, probably due to larger differences in the PBL, which is better resolved at L58 (Figure 3, Figure S3). The microphysics is not as directly related to the cloud fraction, which means interaction with the boundary layer turbulence is important. While these changes are easy to spot, they are not that large, and generally well within some of the tuning which is often done during the model development process."

[Figure]

(updated) Figure 3: Zonal mean climatology from 5-year simulations with CESM2 and CAM6 physics using different dynamical cores and vertical levels.

[Figure]

(new) supplemental Figure S2: Spatial differences between MPAS and FV at L32 for A) shortwave cloud forcing (SWCF), B) longwave cloud forcing (LWCF), C) cloud liquid water path (GGCLDLWP), D) cloud ice water path (TGCLDIWP), E) total cloud (CLDTOT), F) low cloud(CLDLOW), G) high cloud (CLDHGH), H) droplet concentration (CDNUMC), J) 10m wind speed (U10), K) aerosol optical depth 550 nm (AODVIS).

[Figure]

(new) supplemental Figure S3: Similar as Figure S2 but for spatial differences between MPAS and FV at L58.

Line 250-260 – to me the difference maps in Figure S1 highlight that CAM-FV and MPAS are dissimilar, particularly at high altitudes in the tropics and poles (they even have different signs in bias).

We agree and have updated the text to reflect the fact that the biases relative to reanalysis are different between MPAS and FV, but they are not larger in magnitude, and in the refined region of interest, biases are small.

The added text reads: "Analysis of the atmospheric wind and temperature structure (Figure S4 and Figure S5) indicates that SIMA-MPAS compares as well or better to reanalysis winds and thermal structure in the vertical as SIMA-FV, though biases are different and of a different sign in many regions of the middle atmosphere. There are differences in low level wind speed and the subtropical jets between MPAS and FV (Figure S4), driving differences in temperature between them (Figure

S5), particularly in the stratosphere and near the south pole. The stratosphere and free troposphere winds differences are due to slightly different damping and deposition of gravity wave drag forcing. The temperature changes above the surface respond to those wind changes. The near-surface temperature differences (e.g., around Antarctica) also relate to transport of air around topography which is different between MPAS and FV."

Line 265 – Delete "coastal ranges and" (redundant)

Deleted. Thank you.

Line 273-275 – Why is "for about" used? What are spatial textures?

Deleted "for about". Reworded "spatial textures" to "spatial details".

Figure 3 – what aren't a) and b) on the same page?

We have corrected this figure formatting.

Line 280 – Delete one of the ")"…

Deleted. Thank you.

Line 286-287 – I'm not following the reasoning about the "smoother topography". The innermost circle (representing the refinement patch of SIMA-MPAS) in Figure 1a appears to properly encapsulate the entirety of the WUS. Can you provide a difference plot of Figure 2 (and from an observed DEM) for easier comparison (place in Supplemental)? Also, average precipitation rate is low biased (compared with PRISM) throughout the Klamath, Great Basin, and Sierra Nevada too (which is well within the innermost domain of SIMA-MPAS). Could this also be an interannual variability issue with only 5-years of simulation (i.e., atmospheric rivers, etc. may not have made landfall as frequently or in the same location or at the same magnitude as what was observed)?

We have updated the argument in the text that discusses the precipitation biases, which reads: "In terms of biases, SIMA-MPAS 3km overall underestimates the precipitation by about 0.07 mm (bias averaged over the plotted domain), especially over the windward regions, which could relate to the bias in heavy precipitation frequency and/or the discrepancies in ARs landfalling locations and magnitude from what was observed over the five-year (wet-season) simulation statistics"

Given the visualization for the native grid topography (Figure 2), WRF topography is less smooth than SIMA-MPAS over the north and eastern boundary regions as in the plotted domain. For a detailed examination, we have added the difference (new supplementary plot as Figure S4) between SIMA-MPAS and WRF (which is based on the observed DEM data for the terrain), the topo differences over the high terrains (both data remapping at 4km before the differing) tells the SIMA-MPAS has smoothed topography over the transient region of the 3km mesh.

[Figure]

(new) supplemental Figure S1: Topography differences (unit: m) between SIMA-MPAS (3km) and WRF (4km) (both regridded at the same resolution of 4km) over the western US region.

Line 284 – Again, what is "precipitation texture"?

Thank you. We revised this to "spatial details of the precipitation".

Figure 5 – Change "frequency" to "intensity". Also, I think it will be important to compare/contrast with, at least, one other observationally based gridded precipitation product (see earlier comment).

Thanks. Corrected. And, as suggested, we have added the other observationally based gridded precipitation product for comparison as responded earlier.

Line 320-332 – Do the authors have any idea as to why precipitation intensity (particularly in the extremes) is so much lower in SIMA-MPAS and so much higher in WRF? Providing some context for the readers could be useful (and, if possible, a path forward to fixing this).

It is not known why the biases in SIMA-MPAS are smaller than in WRF. One hypothesis that would limit precipitation intensity is that SIMA-MPAS has strict conservation limits for energy and mass throughout the model, which are not present in WRF. This is a subject for future work, but may also be dependent on the specific WRF physics options used. We have updated the text accordingly.

Line 344-346 – Rather than simply stating MG3 "could be a better option", could the authors spell out in more detail the physical intuition as to why this might be the case (over a few sentences)? Do the additional ice phase hydrometeors alter droplet coalescence/droplet size and the intensity of precipitation, seeder-feeder effects in complex terrain can begin to arise, or …?

Thank you for the suggestion. We have added this detail in the text, which reads: "Gettelman et al 2019 found that the addition of rimed ice improved the simulation of precipitation in CESM at 14km resolution with wintertime orographic precipitation, due to altering the timing of precipitation by more correctly representing the pathways for precipitation formation with higher resolved scale vertical velocities."

Figure 6 – rather than changing the color of MG3, could the authors change the stippling of the line and keep the same line color? I do think interannual variability is likely shaping the California results (e.g., not enough atmospheric river landfalls as was observed).

As suggested, we have updated the line style and color for MG3. Further, we added the argument in the text for the interannual variability saying that "We also acknowledge that the interannual variability and the sample size of the ARs could also affect the results of landfalling precipitation."

Line 369 – What are "snowpack statistics features"?

We have rephrased this to "Accumulated snowpack features".

Line 371-375 – This is a confusing set of sentences. Please revise.

Thank you. We thoroughly revised these sentences to "Snowpack characteristics have remained poorly represented in global climate models, lacking high-resolution terrain realization, fine-scale land-atmosphere coupled processes and interactions with snow's complicated thermal and hydrological properties (DeWalle & Rango 2008; Liu et al., 2017; Kapnick et al., 2018). Facing this long-standing issue, we expect that with much improved precipitation features, temperature, and substantially better-resolved complex terrains, snowpack features can be much better represented in CESM."

Line 381 – Change "retrieving" (too colloquial and not sure what is meant)

Thanks. We changed this wording to "simulating".

Line 371-384 – this is a basic analysis of snowpack skill. Could the authors use a similar figure structure as Figure 6/7 to daily dSWE (mm/day) magnitudes across the WUS? This would be a proxy for snowfall rate while still being able to leverage the observationally-based gridded SWE products that, likely, don't provide it.

We have carefully considered this option. Given the focus of this work, we decided to keep the diagnostics we have for the mean snow accumulation over the investigated seasons instead of the daily changes (loss or addition) of snowpack (i.e. snowmelt and snowfall). However, as responded earlier, we did replace the Daymet with the other observationally-based gridded SWE product as suggested.

Line 408-410 – What is meant by this statement? Are they authors trying to setup that future climate change analyses would be justified based on these results?

Thank you. We have deleted this sentence.

Line 419 – Again, please proofread before submitting a manuscript, "For further investigation, we have investigated'…

Thank you. Corrected.

Line 440-448 – This entire paragraph needs to be overhauled. In particular, the first and last sentences.

Thanks. We revised the paragraph to "In Figure 10, we further examine the large-scale moisture flux pattern from the integrated water vapor transport in the set of simulations with and without regional refinement. It can be seen that the spatial pattern of the moisture flux is generally similar between those two sets of experiments, dominated by the zonal winds (see Figure 9). If checking the IVT values along the longitude of 130˚W, the differences (about 3% on average) are quite small along the WUS extent. With the large-scale dynamics and local fine-scale processes well integrated into this nonhydrostatic global climate model, it gives confidence in precipitation reproducing and predicting across the weather and climate scales."

Line 500-502 – Not sure what is meant by this sentence. Please revise.

As previously commented, we have deleted this sentence.

Line 505-510 – Not sure what is meant by this sentence. Please revise.

Thanks. We have revised these sentences to "We focus on multiple-season statistics for model performance. Given the large-scale dynamics and local fine-scale processes well integrated into this nonhydrostatic global climate model, it shows promise in precipitation reproducing and predicting across the weather and climate scales."

Line 454-514 – To provide more added value from this study, I think the authors need to compare some of the biases in SIMA-MPAS to previous studies using MPAS, VR-CESM, etc. (a paragraph or two) to contextualize model performance, especially if those studies also used PRISM, Daymet, etc. The authors frequently use the term "better", "pretty good", "good", etc. and I think these qualitative statements need to be quantified and contrasted a bit more.

Although it is not possible for a direct comparison to previous studies using MPAS and VR-CESM, we have provided some relevant literature review in the introduction section, which reads: "Specifically, Rhoades et al. (2016) found that the VR-CESM framework (with refinement at 0.25° and 0.125° resolutions) can provide much enhanced representation of snowpack properties relative to widely used GCMs (such as CESM-FV 1° and CESM-FV 0.25°) over the California region. Gettelman et al. (2018) found that the variable-resolution CESM-SE simulation (at 0.25°, ~25 km) can produce precipitation intensities similar to the high-resolution, and has higher extreme precipitation frequency than the low-resolution simulation over the Continental United States (CONUS) refinement region, close to observations. … For example, Rauscher et al. (2013) found that tropical precipitation increases with increasing resolution in the CAM-MPAS using aquaplanet simulations."

We have also made adjustments to those qualitative statements throughout the manuscript for further clarification.